# DYNAMIC SPECULATIVE AGENT PLANNING

**Yilin Guan**[1]   **Qingfeng Lan**[2]   **Fei Sun**   **Dujian Ding**[3]   **Devang Acharya**[4]
**Chi Wang**[5]   **William Yang Wang**[6]   **Wenyue Hua**[6]

[1]Johns Hopkins University   [2]University of Alberta   [3]University of British Columbia
[4]Avey Research Center   [5]Google DeepMind   [6]University of California, Santa Barbara
yguan29@jh.edu   wenyuehua@microsoft.com

## ABSTRACT

Despite their remarkable success in complex tasks propelling widespread adoption, large language-model-based agents still face critical deployment challenges due to prohibitive latency and inference costs. While recent work has explored various methods to accelerate inference, existing approaches suffer from significant limitations: they either fail to preserve performance fidelity, require extensive offline training of router modules, or incur excessive operational costs. Moreover, they provide minimal user control over the tradeoff between acceleration and other performance metrics. To address these gaps, we introduce **Dynamic Speculative Planning** (DSP), an asynchronous online reinforcement learning framework that provides lossless acceleration with substantially reduced costs without requiring additional pre-deployment preparation. DSP explicitly optimizes a joint objective balancing end-to-end latency against dollar cost, allowing practitioners to adjust a single parameter that steers the system toward faster responses, cheaper operation, or any point along this continuum. Experiments on two standard agent benchmarks demonstrate that DSP achieves comparable efficiency to the fastest lossless acceleration method while reducing total cost by 30% and unnecessary cost up to 60%. Our code and data are available through https://github.com/guanyilin428/Dynamic-Speculative-Planning.

## 1 INTRODUCTION

LLM-based agents have rapidly moved from controlled demonstrations to real-world deployments, supporting complex multi-step tasks in domains such as autonomous software engineering (Wu et al., 2024), open-world tool use (Shen et al., 2024), and everyday personal assistance (Ge et al., 2024). Although LLM-based agents have achieved widespread adoption, practitioners consistently report substantial end-to-end latency (Hua et al.; Zhang et al.; 2024b; Saha et al., 2025). This latency not only degrades user experience but also limits the feasibility of deploying LLM-based agents in time-sensitive scenarios such as real-time decision support (Cai et al., 2025) or interactive tutoring (Modran et al., 2024; Piro et al., 2024).

Multiple lines of research attempt to address the latency bottlenecks. These include approaches such as context length reduction (Zhang et al.), step parallelization (Hua et al.; Li et al., 2025; Pan et al., 2025), and dual-process thinking that combines fast and deliberate reasoning modes (Saha et al., 2025; Lin et al.; Christakopoulou et al.). While these approaches offer various improvements in efficiency, there are several common critical limitations: (1) **No guarantee on lossless performance** except ISP: Saha et al. (2025) requires training a router between two agents and Zhang et al. requires careful prompt engineering to enable runtime tool-selection, both relying on the quality of the additional component. Pan et al. (2025) requires the model to learn to generate parallelizable steps. (2) **Pre-deployment preparation**: Saha et al. (2025); Pan et al. (2025) require training specialized modules in the system before deployment, adding complexity and overhead to implementation. (3) **Lack of user-controllability** (Kiefer et al., 2017): Most existing approaches neglect user-controlled flexibility in navigating efficiency improvements. This inflexibility becomes particularly problematic in the rapidly evolving LLM ecosystem, where pricing structures, model inference speeds, and model performance change frequently, and organizations have diverse priorities regarding latency versus

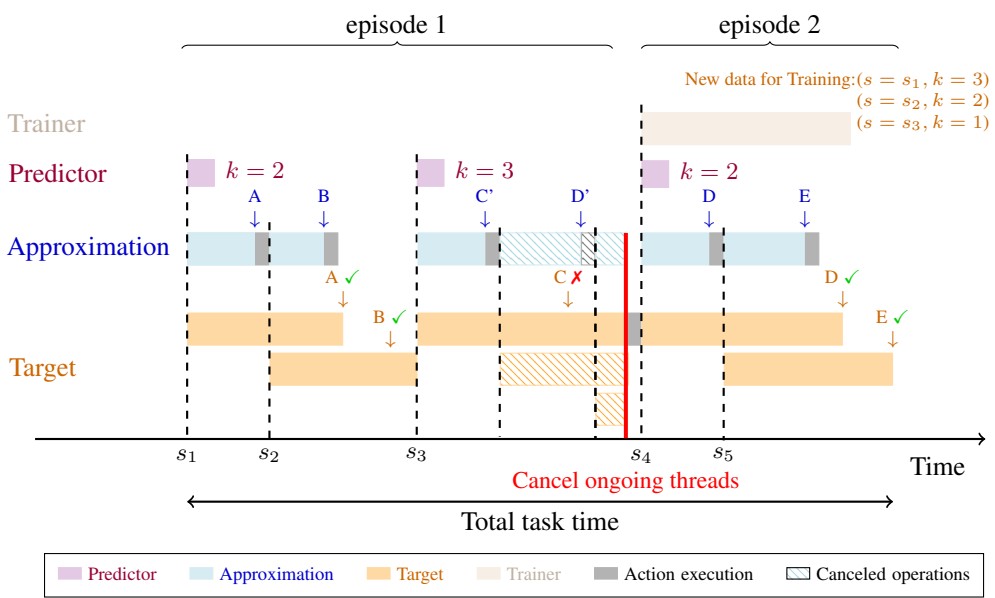

**Figure 1:** Dynamic Speculative Planning with Online Reinforcement Learning. The diagram illustrates how DSP adaptively adjusts speculation steps across planning episodes. (1) **Predictor** dynamically infers step values $k$ from current state $s_i$, while the **Approximation** (**A**) and **Target** (**T**) agents execute parallelly. (2) When mismatches occur (denoted by ✗), ongoing threads are canceled and execution resumes from the last verified step. (3) An asynchronous **Trainer** collects state-step $(s, k)$ pairs from completed episodes, continuously updating the predictor without blocking execution.

operational costs including router training, prompt engineering, or extra cost brought by parallelizing computations.

In this paper, we propose Dynamic Speculative Planning (DSP), a lossless, user-controllable agent planning acceleration framework that requires no pre-deployment preparation. In addition to ensuring lossless agentic performance via speculative strategy (Hua et al.), DSP introduces three core improvements for cost reduction and user-controllability:

First, we **investigate the Pareto frontier** of the latency-cost trade-off in speculative agent planning, determining the optimal cost for any given latency reduction target. This trade-off is governed by the *speculation step*, which refers to the number of future steps that an approximation agent attempts to predict ahead of verification. It determines how aggressively the system parallelizes computation to reduce latency. Our analysis reveals that the commonly adopted fixed-speculation-step approach (Hua et al.; Leviathan et al., 2023) is fundamentally limited: for complex tasks, aggressive speculation produces excessive and redundant agent calls that substantially increase costs, while for simpler tasks, conservative speculation fails to deliver sufficient acceleration. Since optimal speculation step is highly context-dependent and unknown a priori, fixed steps significantly constrain practical utility.

Second, we develop **a lightweight adaptive speculation step predictor** that dynamically determines when to suspend speculation, effectively eliminating unnecessary cost while preserving acceleration benefits. This predictor employs *online reinforcement learning* that requires no external datasets, preprocessing steps, or dedicated pre-deployment phases. Instead, the system learns to optimize speculation step organically as it processes tasks, becoming increasingly efficient over time with zero additional infrastructure costs while ensuring immediate deployment benefits.

Third, based on this dynamic prediction module, we introduce **two mechanisms that modulate the trade-off between latency and cost**: (1) biased step prediction and (2) unbiased step with biased offset, allowing practitioners to precisely calibrate system behavior anywhere along the spectrum from low-cost/low-speed to high-cost/high-speed operation. By providing fine-grained user control over the latency-cost balance, our framework accommodates diverse organizational priorities and adapts to the fluctuating pricing structures and inference speeds of the rapidly evolving LLM ecosystem.

The structure of our system is presented in Figure 1. Our empirical evaluation across two standard agent benchmarks demonstrate that DSP achieves the efficiency of the fastest lossless acceleration methods, while reducing total cost by up to 30%, cutting unnecessary cost by 60%, and preserving comparable acceleration, demonstrating the effectiveness of the framework.

## 2 RELATED WORK

**Latency in LLM-based Agent** LLM-based agents have gained widespread adoption in real-world applications Ge et al. (2024); Wu et al. (2024) due to their strong planning capabilities. However, significant latency issues have been documented across multiple domains: Jaech et al. (2024) reports 24-hour processing windows for o1-based agents in engineering tasks, while Elfleet & Chollet (2024) and Liu et al. (2024a) notes that perceived latency and extended inference time negatively impacts user experience and render LLM agents unsuitable for real-time applications such as virtual reality and gaming.

Several research directions have emerged to address latency challenges. System 1.x Saha et al. (2025) leverages dual-process theory of cognition Evans (2003); Hua & Zhang (2022), dynamically substituting the primary backbone with smaller models when planning subtasks are identified as straightforward, thereby reducing individual inference times. EcoAct Zhang et al. addresses a different source of delay by optimizing prompt length through the selective omission of tool descriptions irrelevant to the current step. Interactive Speculative Planning (ISP) Hua et al. accelerates execution by parallelizing action generation and verification, transforming the inherently autoregressive planning pipeline to non-autoregressive. However, a critical gap exists in the literature: these approaches either compromise performance guarantees to achieve acceleration, require extensive pre-deployment preparation, or incur prohibitive computational costs without providing mechanisms to control latency trade-offs. Our work bridges this gap by **introducing a lossless self-optimizing, zero-setup framework** based on speculative execution that can adaptively balance latency reduction against operational costs, providing the first comprehensive solution for deploying accelerated LLM agents in diverse real-world contexts with varying efficiency requirements.

**Speculative execution** Kocher et al. (2020); Xiao et al. (2020) originated in computer architecture to reduce latency by executing instructions before confirmation, with roll back mechanisms for incorrect speculations. Speculative decoding Zhang et al. (2024a); Leviathan et al. (2023) adapts this principle to LLMs: an efficient draft model predicts multiple future tokens, which a more capable while slow target model verifies in parallel. Tokens that pass verification are committed to the output, while rejected tokens are regenerated, ensuring lossless performance. To improve efficiency, one line of work aligns draft models with target models through online distillation methods Liu et al. (2024b); Zhou et al. (2024); Ouyang et al. (2024); Gui et al. (2024) or training a dependent head Cai et al. (2024); Li et al. (2024a). Another line of work adapts dynamic draft lengths strategies such as Liu et al. (2025); Li et al. (2024b); Mamou et al. (2024). ISP extends this paradigm to agent planning, where an approximation agent ($\mathbf{A}$) generates tentative planning steps while a stronger target agent ($\mathbf{T}$) verifies them asynchronously. While ISP's plug-and-play approach preserves planning performance, its fixed speculation step leads to either excessive costs or insufficient acceleration. Our work addresses this limitation by developing a dynamic speculation system that automatically determines optimal steps through online reinforcement learning, significantly reducing costs while preserving speed. Additionally, we introduce user-controllable mechanisms that expose the latency-cost continuum, enabling precise system calibration based on operational priorities, a capability absent in all previous speculative execution approaches.

## 3 WHY IS SPECULATIVE PLANNING EXPENSIVE?

**Speculative planning** extends the principle of speculative execution to multi-step sequential decision making in LLM-based agents. This approach employs a dual-agent architecture with a "draft-and-verify" paradigm to accelerate planning while preserving solution quality. In this framework, a computationally efficient $\mathbf{A}$ rapidly generates a sequence of candidate actions, while simultaneously the more capable $\mathbf{T}$ (the original planning agent) utilizes the prefix trajectory from the $\mathbf{A}$ to simultaneously generate its own next actions as verification of the proposals of $\mathbf{A}$. This parallel processing architecture enables multiple verification steps to proceed concurrently rather than sequentially. When

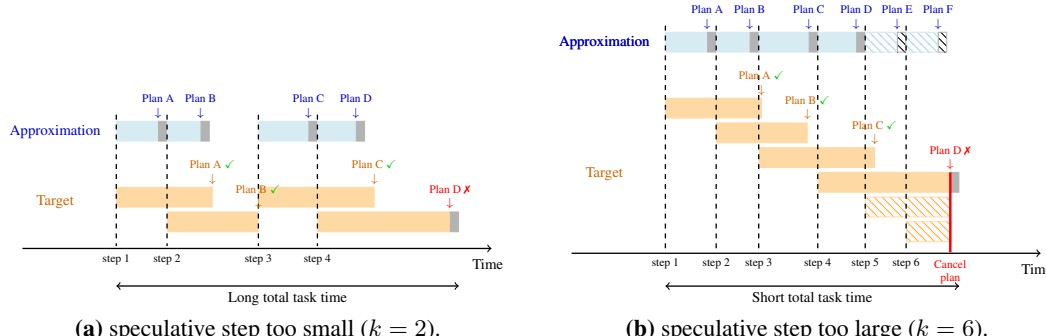

**(a)** speculative step too small ($k = 2$).      **(b)** speculative step too large ($k = 6$).

**Figure 2:** Limitations of fix speculative step approaches. We denote the number of speculative steps as $k$. A small $k$ limits acceleration although it avoids waste, while a large $k$ increases speedup at the cost of redundant computation when mismatches occur (step 5 & 6 is wasted in (b)).

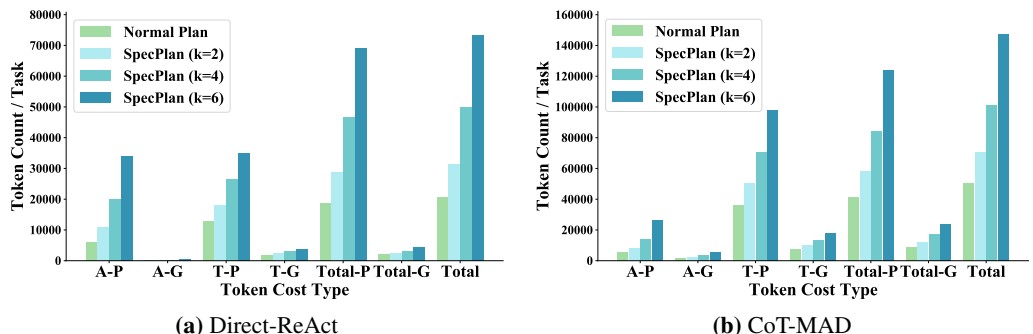

**(a)** Direct-ReAct      **(b)** CoT-MAD

**Figure 3:** Token cost breakdown under speculative planning for different fixed $k$ values on the OpenAGI benchmark, shown for two distinct **A**–**T** configurations: (a) Direct (**A**) with ReAct (**T**), and (b) CoT (**A**) with Multi-Agent Debate (**T**). Each bar contains four segments: **A**'s prompt (A-P) and generation (A-G) tokens, and **T**'s prompt (T-P) and generation (T-G) tokens. These components are also aggregated into total prompt (Total-P)and generation tokens (Total-G), and overall total tokens (Total). In both subfigures, higher values of $k$ result in substantially greater total token costs.

**T** confirms alignment with **A**'s proposals, those actions are immediately committed to the final plan. This verification process proceeds in parallel threads, significantly reducing the end-to-end latency compared to traditional autoregressive planning. Whenever divergence occurs, where **T** proposes a different action than the **A**, the system adopts **T**'s alternative and continues planning from this corrected trajectory. This mechanism ensures that the final execution path always conforms to **T**'s policy while achieve substantial acceleration.

**Speculative Planning Cost Broken-Down**. While speculative execution provides acceleration in a lossless manner, it inevitably introduces additional token consumption. As shown in Figure 2b, this overhead becomes particularly significant when $k$ (the number of speculative steps) is large, motivating the need for a more adaptive planning approach. In this section, we aim to understand where the major overheads arise by systematically breaking down the costs of speculative planning under different fixed-$k$ settings. Redundant costs arise when token consumption does not contribute to the final generated plan or acceleration, which can be divided into four main components: redundant (1) the prompt tokens and (2) the generation tokens consumed by **A**, and (3) the prompt tokens and (4) the generation tokens consumed by **T**. Figure 3 provides a detailed breakdown of token consumption for each component, comparing normal planning against fixed-$k$ speculative planning across different $k$s with 2 speculative planning configurations.

*Approximation agent* incurs excessive wasted token generation when the speculative step $k$ exceeds the number of correct steps the agent can generate sequentially. When a mismatch occurs before reaching $k$, all subsequent generated steps become wasted. This effect intensifies with large-$k$, where frequent divergence at steps smaller than $k$ leads to more discarded tokens. As shown in Figure 3, **A**'s waste primarily stems from prompt tokens rather than generation tokens. This occurs because prompts

contain lengthy environment descriptions, task details, and action histories, while generations are relatively brief, consisting only of actions and possibly some reasoning thoughts.

*Target agent* also incurs excessive wasted token from mismatches between $\mathbf{A}$ and $\mathbf{T}$. These mismatches introduce two additional sources of redundant costs: (1) **Completed T steps**: When mismatches occur, all subsequent completed target steps become invalid, wasting their prompt and generation tokens entirely. (2) **Ongoing target steps**: All subsequent in-progress target steps after a mismatch are immediately canceled. These steps may have already initiated LLM calls, consumed prompt tokens, and generated partial outputs before forced cancellation. This type of waste grows proportionally with $k$ because speculative planning concurrently launches $\mathbf{A}$ and $\mathbf{T}$ at each step. Larger $k$ naturally lead to more target steps running ahead of validation, amplifying potential waste caused by each mismatch.

**Variation of Optimal $k$**   An optimal $k$ is defined as the number of speculative steps from the current state up to the first mismatch, achieving maximal speedup without generating any invalid tokens. The analysis of 312 tasks from the OpenAGI benchmark reveals substantial variability in the optimal speculation step $k$ even within the same task. For each task, we computed the maximum and minimum optimal $k$ across different execution trajectories, as well as the variance and the difference (max–min) of these values: on average across tasks, the maximum optimal $k$ was 3.53 and the minimum was 1.60, with a mean variance of 1.46 and an average range of 1.93. The observed intra-task variability in optimal $k$ suggests that a fixed setting is inadequate for diverse scenarios.

**Implications for Dynamic Speculative Planning**   The cost analysis and heterogeneity of optimal $k$ reveal an inherent tension in speculation step selection: the optimal $k$ varies significantly across different planning stages within a single task and thereby *small $k$* limits acceleration by under-utilizing available concurrency, while *large $k$* frequently trigger excessive token generation which increase costs. Due to the heterogeneous difficulty of step proposals within tasks, no single optimal $k$ exists throughout the plan. Therefore by adaptively modulating $k$ based on contextual factors, such as task characteristics, step complexity, and prediction confidence, a dynamic approach should strategically balance acceleration and efficiency.

*Beyond mere adaptation to task characteristics, a dynamic approach also creates an opportunity for user-controlled efficiency trade-offs.* As model pricing structures and organizational priorities evolve, the optimal balance between latency reduction and cost efficiency varies considerably across deployment contexts. Some applications may prioritize near-instantaneous responses regardless of cost (*i.e.*, critical decision support) or operate in environments where prompt tokens are inexpensive, thus minimizing the financial impact of additional token generation. Others may emphasize cost efficiency with moderate acceleration (*i.e.*, batch processing systems) or operate under constraints where excessive generation tokens or high-priced prompt tokens significantly impact operational budgets. By exposing this latency-cost continuum through configurable parameters, dynamic speculative planning empowers practitioners to precisely calibrate the system.

# 4   DYNAMIC SPECULATIVE PLANNING WITH USER CONTROLLABILITY

Dynamic Speculative Planning begins with a simple intuition: once $\mathbf{A}$ has produced a few actions, it is often possible to forecast how many additional actions it will generate correctly before $\mathbf{T}$ must intervene. Rather than selecting $k$ heuristically or relying on an expensive offline model, we learn to predict it *online* while the system is running. When a mismatch happens, the system logs a training pair (partial trajectory, $k$) for training a small DistilBERT-based regressor whose training cost can be negligible compared with inference cost of a single LLM call. To maintain system responsiveness, the predictor training is done asynchronously: for every newly collected datapoints, an independent thread fine-tunes the predictor to convergence and then preserves the updated checkpoint. The orchestration layer always utilizes the most recent checkpoint, so learning progresses without ever blocking agent planning.

Compared to offline training, our online, multi-threaded training regimen offers two key advantages: (1) *Real-time adaptation*: the predictor is continuously updated with the most recent user interactions, keeping it aligned with the current usage distribution. (2) *Prompt cost savings*: the system begins reducing expenses as soon as new data arrive, without waiting to curate a large offline dataset, which is expensive and inconvenient.

## 4.1 SPECULATIVE-STEP CONTROL AS VALUE PREDICTIONS

As mentioned above, dynamic speculative planning requires effectively predicting the $k$ for each (partial) trajectory and updating the predictor online for continuous adaptation. These requirements make online reinforcement learning a suitable framework. We formalize speculative step prediction as **a state-value prediction problem in reinforcement learning** and apply Temporal-Difference (TD) learning (Sutton & Barto, 2018) to online update this state-value function.

Formally, let $\Delta(X)$ be the space of all probability distributions supported over the set $X$. Consider a Markov decision process (MDP), $M = (\mathcal{S}, \mathcal{A}, \mathrm{P}, p_0, R, \gamma)$, where the state space $\mathcal{S}$ is a set of token sequences, the action space $\mathcal{A}$ is a set of speculative steps, $\mathrm{P} : \mathcal{S} \times \mathcal{A} \to \Delta(\mathcal{S})$ is the transition function, the initial state distribution $p_0$ is a probability distribution over the prompt dataset $Q$ (*i.e.*, $p_0 \in \Delta(Q)$), $R : \mathcal{S} \times \mathcal{A} \to \mathbb{R}$ is the reward function, and $\gamma \in [0, 1]$ is the discount factor.

An approximation agent $\mathbf{A}$ interacts with the MDP based on a policy $\pi : \mathcal{S} \to \Delta(\mathcal{A})$. Specifically, for each episode, the agent starts from a prompt $s_0 \sim p_0(\cdot)$. At each time-step $t$, it observes the state $s_t \in \mathcal{S}$, takes an action (*i.e.*, a speculative step) $a_t \sim \pi(\cdot|s_t)$, transitions to a new state $s_{t+1}$ and receives a scalar reward $r_{t+1} = R(s_t, a_t)$. *The episode ends when an approximation-target mismatch is met.* In our case, $s_{t+1}$ is simply a concatenation of the token sequences of $s_t$ and $a_t$, *i.e.*, $s_{t+1} = s_t | a_t$. We also set $\gamma = 1$. After an episode ends, we get a complete trajectory and it is formally defined as $\tau = (s_0, a_0, r_1, s_1, \dots, s_T)$, where $T$ is the final time-step. Define return $G_t$ over $\tau$ as the total (discounted) reward from time-step $t$:

$$G_t = \sum_{i=t}^{T-1} \gamma^{i-t} R(s_i, a_i).$$

A state-value function is defined as the expected return under policy $\pi$:

$$V_\pi(s) = \mathbb{E}_\pi[G_t | s_t = s].$$

The reward function is designed so that the return $G_t$ corresponds to the number of speculative steps at time-step $t$. Specifically, if the speculative step $a_t$ (generated by $\mathbf{A}$) matches the one generated by $\mathbf{T}$, we set $R(s_t, a_t) = 1$; otherwise, $\mathbf{A}$ gets zero reward and the episode ends immediately. Under this reward setting, the state-value $V_\pi(s)$ is essentially the expected number of speculative steps of state $s$.

In practice, $V_\pi$ is usually approximated by a neural network $V_\theta$, where $\theta$ denotes the network weights. To learn a good approximation of $V_\pi$, we apply TD learning and minimize the following loss function:

$$L_\theta = \mathbb{E}_{\tau \sim \pi}[(G_t^\lambda - V_\theta(s_t))^2], \tag{1}$$

where $G_t^\lambda$ is known as $\lambda$-return (Sutton & Barto, 2018). Formally, it is defined as

$$G_t^\lambda = (1-\lambda) \sum_{n=1}^{T-t-1} \lambda^{n-1} G_{t:t+n} + \lambda^{T-t-1} G_t, \tag{2}$$

where $G_{t:t+n} = r_{t+1} + \gamma r_{t+2} + \cdots + \gamma^{n-1} r_{t+n} + \gamma^n V_\theta(s_{t+n})$ and $\lambda \in [0, 1]$. The usage of $\lambda$-return offers a flexible way to achieve a better bias-variance tradeoff by interpolating between Monte Carlo return $G_t$ (when $\lambda = 1$) and 1-step TD target $r_{t+1} + \gamma V_\theta(s_{t+1})$ (when $\lambda = 0$). It is worth noting that when $\lambda = 1$ and $\gamma = 1$, TD learning in this context reduces to online supervised learning since $G_t^1 = G_t = k$, thus our learning framework is a more generalized framework and achieves superior performance compared to purely supervised training (detailed in Section A.8).

**Prediction Decoupled Data Collection**. To ensure unbiased data collection, DSP logs pairs $(s_t, G_t^\lambda)$ only when $\mathbf{A}$ actually fails, rather than when the speculative budget is exhausted. Consider an under-speculation scenario in which $\mathbf{A}$ can generate $k_1$ consecutive valid actions, but the predictor predicts $k_2$ ($k_2 < k_1$) actions. If training data are collected only for $k_2$ steps, we would underestimate and gradually bias the predictor toward conservative strategies. To eliminate this bias, DSP records training samples at the actual failure point ($k_1$). This ensures that *prediction errors affect only system cost and efficiency without biasing the training data* , thereby eliminating the need to counteract systematic underestimation.

**State-Value Estimation Formulation**. DSP frames the speculation problem as value prediction, rather than policy optimization. In our case, both agents ($\mathbf{A}$ and $\mathbf{T}$) are fixed, where $\mathbf{A}$ controls the policy $\pi$ and $\mathbf{T}$ is part of the environment. Moreover, the inherent nature of speculative planning guarantees that the prediction of $k$ values does not alter the distribution of generated trajectories. Thus, DSP curates data incrementally only to estimate the expected returns under the fixed policy, *avoiding the stability issues* caused by policy optimization.

## 4.2 MULTI-THREAD ONLINE LEARNING

To minimize inference latency and update overhead, our system adopts a fully asynchronous learning framework that decouples predictor inference and training from the core execution pipeline: (1) **asynchronous inference for parallel execution**: To mitigate the impact of predictor latency on system throughput, we perform speculative step prediction asynchronously at the start of each episode. The predictor runs concurrently with $\mathbf{A}$'s planning step, and typically completes before $\mathbf{A}$'s current step finishes. This parallelism effectively hides prediction latency. (2) **asynchronous training for continuous adaptation**: To adapt to evolving task distributions, the predictor is updated online via a decoupled training-inference architecture. We maintain two models: a *training* predictor that undergoes continuous updates, and an *inference* predictor that serves all runtime inference requests. Upon completion of each speculative episode, an asynchronous thread is launched independently of the core planning pipeline to train the predictor. Once the training finishes, updated weights are immediately synchronized to the in-memory inference predictor.

## 4.3 CONTROLLING THE COST-LATENCY TRADE-OFF

In speculative planning, the balance between cost and latency is intrinsically tied to the predicted step $k$. Thus, by adjusting the estimation bias of the step $k$, we allow users to control the trade-off between cost and latency according to their specific needs. There are in general two approaches for adjusting the speculative step $k$: one implemented during predictor training through modified loss functions, and another applied at inference time through post-prediction adjustment parameters.

**Biased $k$ Prediction**. One method to flexibly modulate the estimation bias of step values is to implement expectile regression (Kostrikov et al., 2022) during training. This approach applies asymmetric penalties that systematically shift the predicted values in a desired direction, allowing for principled control over the bias-variance tradeoff in step prediction. Essentially, expectile regression generalizes ordinary least square (OLS) regression by replacing (symmetric) mean squared error (MSE) with an asymmetric MSE: $\mathbb{E}_{(x,y)\sim\mathcal{D}}[L_2^\tau(y - f(x))]$ where $L_2^\tau(u) = |\tau - \mathbf{1}(u < 0)|u^2$ and $\tau \in (0, 1)$ determines which expectile to estimate. The loss reduces to MSE when $\tau = 0.5$.

In DSP, we apply expectile regression to train the step predictor $V_\theta(s)$, by simply replacing MSE in Equation (1) with the asymmetric MSE: $L_\theta = \mathbb{E}_{\tau\sim\pi}[L_2^\tau(G_t^\lambda - V_\theta(s_t))]$. By tuning the value of $\tau$, we can flexibly balance cost and latency in dynamic speculative planning. When $\tau > 0.5$, a negative prediction error ($G_t^\lambda - V_\theta(s_t) < 0$) incurs a lower penalty, causing $V_\theta(s)$ to overestimate $G_t^\lambda$, which leads to faster execution but higher cost. Conversely, when $\tau < 0.5$, $V_\theta(s)$ tends to underestimate $G_t^\lambda$, resulting in lower cost but increased execution latency.

$k$ **with Biased Offset**. Another way to flexibly control the bias is to directly add or minus some offset to the unbiased predicted $k$ without retraining the predictor. This method offers a straightforward and convenient mechanism for implementing user-controlled preferences in the latency-cost tradeoff. After the predictor generates its initial unbiased estimate $\hat{k}$, we apply a user-specified bias parameter $\beta \in \mathbb{N}$ to obtain the final step value: $k = \max(1, \hat{k} + \beta)$. Positive values of $\beta$ shift predictions toward more aggressive speculation while negative values produce more conservative predictions. The maximum function ensures that the final step is at least 1, maintaining a valid speculation step.

## 5 MAIN EXPERIMENT

This section describes the experimental setup and evaluation results. We assess our approach on two benchmarks: OpenAGI (Ge et al., 2024) which focuses on complex decision-making tasks that require reasoning over both vision and language modalities including 312 multi-step tasks, and TravelPlanner (Xie et al., 2024) which targets travel planning scenarios with 180 tasks. We evaluate the performance

**Table 1:** Dynamic $k$ vs. Fixed $k$ Strategies on **OpenAGI** benchmark with **GPT** backbone.

| Mode | Direct-ReAct (Setting 1) | | | | | | CoT-MAD (Setting 2) | | | | | |
|---|---|---|---|---|---|---|---|---|---|---|---|---|
| | T ($\times$) | P ($\times$) | G ($\times$) | Cost ($\times$) | $\overline{MC}$ | $\overline{K}$ | T ($\times$) | P ($\times$) | G ($\times$) | Cost ($\times$) | $\overline{MC}$ | $\overline{K}$ |
| Fix (k = 2) | 1.00 | 1.00 | 1.00 | 1.00 | 3.00 | 2.00 | 1.00 | 1.00 | 1.00 | 1.00 | 3.00 | 2.00 |
| Fix (k = 4) | 0.92 | 1.56 | 1.23 | 1.41 | 4.95 | 4.00 | 0.83 | 1.41 | 1.38 | 1.37 | 4.97 | 4.00 |
| Fix (k = 6) | **0.90** | 2.24 | 1.57 | 1.92 | 6.32 | 6.00 | 0.81 | 1.87 | 1.75 | 1.77 | 6.44 | 6.00 |
| Dyn. ($\tau$=0.5) | 1.01 | 0.87 | 0.91 | **0.88** | 4.33 | 1.78 | 0.98 | 0.95 | 0.95 | **0.95** | 4.44 | 2.10 |
| Dyn. ($\tau$=0.8) | 0.97 | 0.97 | 0.98 | 0.97 | 4.41 | 2.33 | 0.89 | 1.04 | 1.04 | 1.04 | 4.73 | 2.67 |
| Dyn. ($\tau$=0.9) | 0.95 | 1.04 | 1.00 | 1.02 | 4.56 | 2.47 | 0.86 | 1.12 | 1.11 | 1.11 | 5.22 | 3.03 |
| Dyn. ($\tau$=0.95) | 0.94 | 1.12 | 1.07 | 1.09 | 4.75 | 2.88 | 0.85 | 1.26 | 1.25 | 1.24 | 5.64 | 3.58 |
| Dyn. ($\tau$=0.99) | 0.91 | 1.32 | 1.19 | 1.25 | 5.24 | 3.59 | 0.82 | 1.41 | 1.37 | 1.37 | 5.87 | 4.06 |
| Dyn. (offset=1) | 0.92 | 1.10 | 1.06 | 1.08 | 4.86 | 2.75 | 0.88 | 1.11 | 1.09 | 1.10 | 4.95 | 2.80 |
| Dyn. (offset=2) | **0.90** | 1.33 | 1.15 | 1.25 | 5.13 | 3.47 | **0.80** | 1.29 | 1.27 | 1.27 | 5.60 | 3.99 |

of our dynamic speculative planning framework and compare it to a set of baselines.We also provide a comprehensive evaluation of the **predictor's performance** by evaluating both its prediction accuracy and warm-up speed (experiment details in Section A.6).

**Baselines** are chosen to demonstrate the advantages of our approach across three multiple dimensions. (1) We compare against ***fixed-$k$ speculative planning*** ($k \in \{2, 4, 6\}$) to show the effectiveness of dynamic $k$ selection over static approaches that ignore environment characteristics and task requirements. (2)We include a non-contextual dynamic optimization baseline using ***Bayesian Optimization*** to demonstrate the necessity of our sophisticated neural predictor (detailed in Section A.7). (3) Additionally, we evaluate ***supervised fine-tuning*** approaches within our framework to highlight the advantages of our RL formulation (detailed in Section A.8).

**The speculative step predictor** uses DistilBERT for its efficiency and expressive capacity in forecasting the speculation step $k$ from a given state. The model is fine-tuned using expectile regression to learn the bias and adaptively adjust its predictions based on the collected data. The model is optimized using AdamW with a learning rate of $1 \times 10^{-5}$, batch size 16. Each training process iterates over 3 epochs on a randomly sampled replay buffer (size 2,500). For multi-step return estimation, we adopt $\lambda$-return blending ($\lambda = 0.95$) to balance bias-variance trade-offs during reward propagation.

**LLM backbones for agents** from the GPT and DeepSeek families are used in DSP. Below are their cost profiles: (1) *GPT-4.1-mini* costs \$0.40/\$1.60 (prompt/generation per million tokens). (2) *DeepSeek*: We use DeepSeek-chat as **A**'s backbone and DeepSeek-reasoner as **T**'s backbone, with respective costs of \$0.27/\$1.10 and \$0.55/\$2.19.

**Four DSP configurations** are utilized for experiments: (1) *Direct-Generation*: **A** generates the action directly for each step in a single LLM call, without any intermediate reasoning. (2) *Chain-of-Thought (CoT)*: **A** first reasons about the current step and then generates the action based on the reasoning, all within a single LLM call (Wei et al., 2022). (3) *ReAct*: **T** first deliberates on the action to take, then generates the action through two distinct LLM calls (Yao et al., 2023). (4) *Multi-agent Debate (MAD)*: two target agents debate in two rounds, with each agent proposing and refining actions.

**Evaluation Metrics** are designed along three critical dimensions to assess speculative planning (SP) systems: *latency reduction*, *resource overhead*, and *concurrency dynamics*. This triad captures the trade-offs in real-world deployment, where effective systems must accelerate task completion while respecting resource constraints and API concurrency limits. For *latency reduction* and *resource overhead*, we adopt fixed $k = 2$ as the baseline configuration for evaluating various speculation strategies, as it represents the minimal non-trivial setting that offers meaningful acceleration with minimal resource cost: on the OpenAGI benchmark, the fixed $k = 2$ configuration typically provides 15%-25% time reduction compared to sequential planning. Efficiency and economy metrics are reported as ratios relative to the fixed $k = 2$ baseline. T($\times$), P($\times$), G($\times$), and Cost($\times$) represent relative latency, prompt token usage, generation token usage, and monetary cost, respectively. $\overline{MC}$ denotes average peak concurrency (maximum parallel LLM calls), and $\overline{K}$ is the average predicted step length per episode. (Detailed definition in Section A.2)

## 5.1 MAIN EXPERIMENT RESULT

As shown in Table 1, Table 2, Section 5.1.1, and Figure 7, our experiments validate that DSP achieves superior Pareto efficiency compared to fixed-k baselines. Key findings include:

**Table 2:** Dynamic $k$ vs. Fixed $k$ Strategies on **OpenAGI** benchmark with **DeepSeek** backbone.

| Mode | Direct-ReAct (Setting 3) | | | | | | CoT-MAD (Setting 4) | | | | | |
|---|---|---|---|---|---|---|---|---|---|---|---|---|
| | T (×) | P (×) | G (×) | Cost (×) | $\overline{MC}$ | $\overline{K}$ | T (×) | P (×) | G (×) | Cost (×) | $\overline{MC}$ | $\overline{K}$ |
| Fix (k = 2) | 1.00 | 1.00 | 1.00 | 1.00 | 3.00 | 2.00 | 1.00 | 1.00 | 1.00 | 1.00 | 3.00 | 2.00 |
| Fix (k = 4) | 0.89 | 1.58 | 1.50 | 1.55 | 4.92 | 4.00 | 0.89 | 1.51 | 1.47 | 1.49 | 4.94 | 4.00 |
| Fix (k = 6) | **0.87** | 2.04 | 1.86 | 1.98 | 6.33 | 6.00 | **0.87** | 1.86 | 1.80 | 1.83 | 6.59 | 6.00 |
| Dyn. ($\tau$=0.5) | 1.04 | 0.93 | 0.94 | **0.93** | 3.89 | 1.69 | 1.07 | 0.94 | 0.94 | **0.94** | 3.78 | 1.54 |
| Dyn. ($\tau$=0.8) | 0.98 | 1.02 | 1.02 | 1.02 | 4.20 | 2.08 | 0.97 | 1.08 | 1.08 | 1.08 | 4.57 | 2.31 |
| Dyn. ($\tau$=0.9) | 0.95 | 1.12 | 1.12 | 1.12 | 4.45 | 2.58 | 0.93 | 1.16 | 1.15 | 1.16 | 4.69 | 2.65 |
| Dyn. ($\tau$=0.95) | 0.92 | 1.17 | 1.19 | 1.18 | 4.78 | 2.83 | 0.90 | 1.24 | 1.22 | 1.23 | 5.12 | 3.12 |
| Dyn. ($\tau$=0.99) | 0.88 | 1.42 | 1.40 | 1.41 | 5.46 | 3.85 | 0.88 | 1.39 | 1.36 | 1.38 | 5.29 | 3.75 |
| Dyn. (offset=1) | 0.93 | 1.11 | 1.12 | 1.11 | 4.39 | 2.58 | 0.91 | 1.15 | 1.15 | 1.15 | 4.48 | 2.80 |
| Dyn. (offset=2) | **0.87** | 1.35 | 1.34 | 1.34 | 4.99 | 3.53 | 0.88 | 1.36 | 1.33 | 1.35 | 5.38 | 3.85 |

| Mode | Direct-ReAct (Setting 1) | | | | | | CoT-MAD (Setting 2) | | | | | |
|---|---|---|---|---|---|---|---|---|---|---|---|---|
| | T (×) | P (×) | G (×) | Cost (×) | $\overline{MC}$ | $\overline{K}$ | T (×) | P (×) | G (×) | Cost (×) | $\overline{MC}$ | $\overline{K}$ |
| Fix (k = 2) | 1.00 | 1.00 | 1.00 | 1.00 | 3.00 | 2.00 | 1.00 | 1.00 | 1.00 | 1.00 | 3.00 | 2.00 |
| Fix (k = 4) | 0.89 | 1.95 | 1.39 | 1.95 | 5.00 | 4.00 | **0.96** | 1.48 | 1.22 | 1.46 | 4.62 | 4.00 |
| Fix (k = 6) | **0.87** | 2.86 | 1.67 | 2.91 | 6.99 | 6.00 | 0.97 | 1.67 | 1.25 | 1.64 | 5.37 | 6.00 |
| Dyn. ($\tau$=0.5) | 1.05 | 0.81 | 0.92 | **0.80** | 4.28 | 1.46 | 1.05 | 0.85 | 0.88 | **0.84** | 4.09 | 1.30 |
| Dyn. ($\tau$=0.8) | 0.99 | 0.95 | 0.97 | 0.93 | 4.60 | 1.88 | 1.05 | 0.86 | 0.89 | 0.85 | 4.00 | 1.42 |
| Dyn. ($\tau$=0.9) | 0.95 | 1.06 | 1.03 | 1.04 | 4.96 | 2.21 | 1.01 | 0.99 | 0.96 | 0.98 | 4.03 | 1.79 |
| Dyn. ($\tau$=0.95) | 0.93 | 1.20 | 1.09 | 1.18 | 5.19 | 2.55 | 0.98 | 1.02 | 0.97 | 1.01 | 4.00 | 2.10 |
| Dyn. ($\tau$=0.99) | 0.92 | 1.49 | 1.20 | 1.48 | 5.32 | 3.10 | **0.96** | 1.25 | 1.13 | 1.24 | 4.58 | 2.94 |
| Dyn. (offset=1) | 1.03 | 0.96 | 0.97 | 0.94 | 4.31 | 2.30 | 0.99 | 1.05 | 1.01 | 1.04 | 4.08 | 2.16 |
| Dyn. (offset=2) | 0.99 | 1.16 | 1.04 | 1.12 | 4.91 | 3.22 | 0.98 | 1.33 | 1.16 | 1.32 | 4.33 | 2.98 |

### 5.1.1 COST-EFFICIENCY DOMINANCE

DSP demonstrates **strict Pareto dominance** through two operational modes:

**(1) Equivalent Latency, Lower Cost**. *On the OpenAGI benchmark*:

- In **setting 1**, Dyn. ($\tau = 0.99$) and Dyn. (offset = 2) achieves comparable time reduction to Fix ($k = 6$) with a **34.9%** (1.25× compared with 1.92× as in Table 1) reduction in cost. Dyn. (offset = 1) saved **23.4%** total cost compared to Fix ($k = 4$) with comparable latency reduction. Compared to Fix ($k = 2$), Dyn. ($\tau = 0.5$) achieves a similar latency reduction and a substantial 12% decrease in token cost.
- In **setting 2**: Dyn. ($\tau = 0.95$) achieves comparable latency reduction to Fix ($k = 4$) while saving 9.49% cost. Dyn. (offset = 2) yields a marginally better acceleration than Fix ($k = 6$), and significantly reduces token cost by **28.25%**.
- In **setting 3**: Dyn. ($\tau = 0.99$) and Dyn. (offset = 2) saves 6.78% and 10.67 % cost compared to Fix ($k = 4$) with a marginal acceleration improvement.
- In **setting 4**: Dyn. ($\tau = 0.99$) saves 7.38% cost compared to Fix ($k = 4$) with comparable latency reduction. *On the Travel Planner benchmark*: Dyn. ($\tau = 0.8$) achieves a 7% reduction in cost in **setting 1**, along with modest latency improvements compared to Fix ($k = 2$).

**(2) Equivalent Cost, Faster Execution**. *On the OpenAGI benchmark*:

- In **setting 1**, Dyn. ($\tau = 0.9$) delivers 5% relative acceleration over Fix ($k = 2$), only consuming 2% more cost.
- In **setting 2**, Dyn. ($\tau = 0.5$) provides 8% better latency reduction while consuming 5% less cost than Fix ($k = 2$).
- In **setting 4**, Dyn. (offset = 2) requires 9.52% fewer cost to surpass Fix ($k = 4$).

DSP consistently outperforms all fixed-$k$ baselines in terms of cost-effective acceleration in all the settings, demonstrating its adaptability and generality. We provide a comprehensive analysis of unnecessary cost reduction and time acceleration relative to sequential planning in Section A.3, which

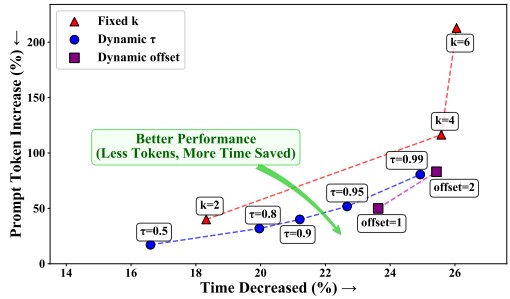 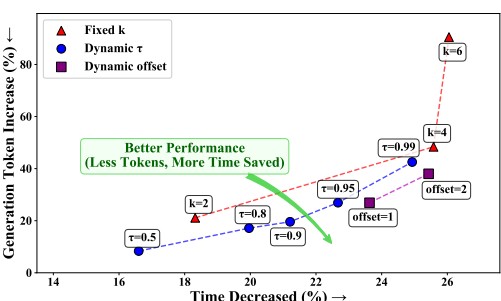

**(a)** Prompt Token Increase vs Time Decrease     **(b)** Generation Token Increase vs Time Decrease

**Figure 4:** Performance trade-off between acceleration and token increase between Fix-$k$, Dynamic-$\tau$ and Dynamic-offset SP. (a) Prompt token and (b) Generation token increase vs. latency reduction, both relative to a normal-plan baseline. Detailed metric definitions and corresponding performance analysis are provided in Section A.2 and Section A.3.

also demonstrates DSP's ability to effectively identify and exploit parallelism opportunities in a variety of reasoning pipelines. We also provide wall clock time analysis in Section A.12, demonstrating that DSP quickly stabilizes near the optimal latency-cost region and achieves Pareto-frontier fast.

## 6 CONCLUSION

We presented Dynamic Speculative Planning (DSP), a method that addresses fixed-step speculative execution limitations in LLM-based agents through lightweight online reinforcement learning requiring no pre-deployment preparation. Our evaluation demonstrates DSP reduces redundant computational overhead by up to 60% and overall cost by 30% while maintaining acceleration benefits, with user-controllable parameters enabling precise calibration of latency-cost tradeoffs. This advancement improves the viability of deploying sophisticated agents in latency-sensitive real-world applications.

## 7 REPRODUCIBILITY STATEMENT

We provide our implementation at `https://github.com/guanyilin428/Dynamic-Speculative-Planning.git`. The repository includes a detailed README file with step-by-step instructions to reproduce our experiments. All model hyperparameters and training settings are clearly stated in the experimental setup section (see Section 5).

## 8 ETHICS STATEMENT

This work does not involve human subjects, sensitive personal data, or information that could raise privacy concerns. The datasets and benchmarks we use are publicly available, and our methods do not introduce discrimination or unfair bias. We believe our research poses no foreseeable ethical risks and adheres to the ICLR Code of Ethics.

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

# A APPENDIX

## A.1 LLM USAGE

Large Language Models were used in this work solely for grammatical correction and language editing purposes.

## A.2 EVALUATION METRICS

**1. Time Decrease ($\Delta$T)** Quantifies the relative latency reduction when using speculative planning. Specifically, for each task, we calculate the total time taken to complete the task under SP ($T^{\text{SP}}$) and sequential planning ($T^{\text{seq}}$). The percentage time saving is then calculated as:

$$\Delta\text{Time} = \frac{1}{N} \sum_{i=1}^{N} \left( 1 - \frac{T_i^{\text{SP}}}{T_i^{\text{seq}}} \right) \times 100\%$$

where $N$ is the total number of tasks. Positive values indicate time savings.

**2. Cost and Token Increase** Token increase captures the *redundant* resources consumed by speculative planning as a result of incorrect speculation. While the primary goal of speculative planning is to reduce latency and enhance efficiency, it still incurs a minimum cost. Even with perfect accuracy in speculative planning (*i.e.*, the approximation agent makes correct predictions for every step), the cost must at least include the *baseline cost* — the minimal overhead associated with sequential planning of both the target and approximation agents. Thus, we define the baseline for token increase as the sum of the planning costs for the target agent and approximation agent operating sequentially, which represents the least amount of resources required for completing the task with speculation. Any increase in token usage beyond this baseline reflects the additional overhead introduced by speculative failures, capturing the waste generated from incorrect predictions by the approximation agent.

Since the cost per token can differ for *prompt tokens* and *generation tokens*, we separately evaluate these components and also provide a combined metric for *total cost increase*.

**Prompt Token Increase ($\Delta P$)** This measures the increase in prompt tokens due to redundant prompt usage in SP. Let $p_i^{\text{seq}} = p_i^{\text{target}} + p_i^{\text{approx}}$ be the baseline prompt tokens (the sum of target and approximation prompts under sequential execution), and $p_i^{\text{SP}}$ be the prompt tokens used under SP (parallel execution). The prompt token increase is calculated as:

$$\Delta P = \frac{1}{N} \sum_{i=1}^{N} \left( \frac{p_i^{\text{SP}}}{p_i^{\text{seq}}} - 1 \right) \times 100\% \tag{3}$$

**Generation Token Increase ($\Delta G$)** This metric captures the increase in generation tokens caused by incorrect speculation. Let $g_i^{\text{seq}} = g_i^{\text{target}} + g_i^{\text{approx}}$ represent the baseline generation tokens (the sum of target and approximation generations under sequential execution), and $g_i^{\text{SP}}$ the generation tokens used under SP. The generation token increase is calculated as:

$$\Delta G = \frac{1}{N} \sum_{i=1}^{N} \left( \frac{g_i^{\text{SP}}}{g_i^{\text{seq}}} - 1 \right) \times 100\% \tag{4}$$

**Total Cost Increase ($\Delta$Cost)** This metric quantifies the relative increase in cost between speculative planning (SP) and sequential planning (seq) strategies. Specifically, we define the total cost increase as:

$$\Delta\text{Cost} = \frac{1}{N} \sum_{i=1}^{N} \left( \frac{PC_i^{\text{SP}} + GC_i^{\text{SP}}}{PC_i^{\text{seq}} + GC_i^{\text{seq}}} - 1 \right) \times 100\% \tag{5}$$

where $PC_i^{\text{SP}}$ and $PC_i^{\text{seq}}$ represent the prompt costs of task $i$ for SP and sequential strategies, and $GC_i^{\text{SP}}$ and $GC_i^{\text{seq}}$ represent the corresponding generation costs. These costs are computed as follows:

$$PC_i^{\text{SP}} = ap_i^{\text{SP}} \cdot \text{cost}_{\text{ap}} + tp_i^{\text{SP}} \cdot \text{cost}_{\text{tp}} \tag{6}$$

$$PC_i^{\text{seq}} = ap_i^{\text{seq}} \cdot \text{cost}_{\text{ap}} + tp_i^{\text{seq}} \cdot \text{cost}_{\text{tp}} \tag{7}$$

$$GC_i^{\text{SP}} = ag_i^{\text{SP}} \cdot \text{cost}_{\text{ag}} + tg_i^{\text{SP}} \cdot \text{cost}_{\text{tg}} \tag{8}$$

$$GC_i^{\text{seq}} = ag_i^{\text{seq}} \cdot \text{cost}_{\text{ag}} + tg_i^{\text{seq}} \cdot \text{cost}_{\text{tg}} \tag{9}$$

Here, $ap_i^{\text{SP}}$ and $tp_i^{\text{SP}}$ represent the number of prompt tokens for the approximation and target in the SP strategy at task $i$, and $ap_i^{\text{seq}}$ and $tp_i^{\text{seq}}$ represent the number of prompt tokens for the approximation and target in the sequential planning at task $i$. Similarly, $ag_i^{\text{SP}}$, $tg_i^{\text{SP}}$, $ag_i^{\text{seq}}$ and $tg_i^{\text{seq}}$ represent the corresponding number of generation tokens. For cost, $\text{cost}_{\text{ap}}$ and $\text{cost}_{\text{tp}}$ are the costs for the approximation and target API for prompt tokens, while $\text{cost}_{\text{ag}}$ and $\text{cost}_{\text{tg}}$ are the corresponding costs for generation tokens.

By computing the total cost increase, we can quantify the trade-off between speculative planning and sequential planning in terms of the associated computational costs, helping to assess the efficiency of the proposed approach.

**3. Absolute Time Ratio ($\text{T}(\times)$)** This metric quantifies the ratio of the absolute time in the current configuration compared to the baseline configuration (fix $k = 2$), as it represents the minimal non-trivial setting that offers meaningful acceleration with minimal resource cost. It is calculated as:

$$\text{T}(\times) = \frac{T^{\text{current}}}{T^{\text{k=2}}}$$

**4. Absolute Cost and Token Ratio** This metric measures the ratio of the absolute token usage and cost in the current configuration compared to the baseline configuration (fix $k = 2$).

**Absolute Prompt Token Ratio ($\text{P}(\times)$)**

$$\text{P}(\times) = \frac{p_i^{\text{current}}}{p_i^{\text{k=2}}}$$

**Absolute Generation Token Ratio ($\text{G}(\times)$)**

$$\text{G}(\times) = \frac{g_i^{\text{current}}}{g_i^{\text{k=2}}}$$

**Absolute Cost Ratio ($\text{Cost}(\times)$)**

$$\text{Cost}(\times) = \frac{C^{\text{current}}}{C^{\text{k=2}}}$$

**5. Concurrency Dynamics** Evaluating the concurrency behavior of speculative planning (SP) is essential for understanding its real-world deployability. Modern systems often impose limits on simultaneous API requests due to infrastructure constraints or cost-efficiency considerations. Excessive concurrency can lead to API throttling, degraded performance, or instability, making it critical to monitor and control parallel request patterns. To quantify this behavior, we introduce two metrics.

**Average Max Concurrent API Calls ($\overline{MC}$)** For each task $i$, $MC_i$ denotes the peak number of overlapping API requests during speculative execution. $\overline{MC}$ is calculated as:

$$\overline{MC} = \frac{1}{N} \sum_{i=1}^{N} MC_i \tag{10}$$

where $N$ is the number of tasks. This metric reflects how aggressively a system exploits parallelism and whether it respects concurrency constraints under varying speculative policies.

**Table 3:** Performance Comparison of Dynamic $k$ vs. Fixed $k$ Speculative Planning on **OpenAGI** benchmark with **GPT** backbone.

| Mode | Direct-ReACT (Setting 1) | | | | | | CoT-MAD (Setting 2) | | | | | |
|---|---|---|---|---|---|---|---|---|---|---|---|---|
| | $\Delta$T (%) | $\Delta$P (%) | $\Delta$G (%) | $\Delta$Cost (%) | $\overline{MC}$ | $\overline{K}$ | $\Delta$T (%) | $\Delta$P (%) | $\Delta$G (%) | $\Delta$Cost (%) | $\overline{MC}$ | $\overline{K}$ |
| Fix (k=2) | 18.00 | 39.47 | 20.71 | 33.94 | 3.00 | 2.00 | 22.35 | 31.57 | 27.78 | 29.81 | 3.00 | 2.00 |
| Fix (k=4) | 25.06 | 114.94 | 47.88 | 94.89 | 4.95 | 4.00 | 35.62 | 83.78 | 74.46 | 79.38 | 4.97 | 4.00 |
| Fix (k=6) | **25.88** | 212.93 | 90.41 | 176.25 | 6.32 | 6.00 | **37.21** | 145.42 | 122.01 | 134.38 | 6.44 | 6.00 |
| Dyn. ($\tau$=0.5) | 15.68 | 16.40 | 8.23 | **14.04** | 4.33 | 1.78 | 23.12 | 21.20 | 17.83 | **19.63** | 4.44 | 2.10 |
| Dyn. ($\tau$=0.8) | 19.43 | 30.98 | 17.05 | 26.96 | 4.41 | 2.33 | 30.09 | 33.51 | 29.41 | 31.61 | 4.73 | 2.67 |
| Dyn. ($\tau$=0.9) | 20.94 | 39.83 | 19.60 | 34.00 | 4.56 | 2.47 | 32.86 | 43.35 | 38.18 | 40.95 | 5.22 | 3.03 |
| Dyn. ($\tau$=0.95) | 22.31 | 51.36 | 27.00 | 44.22 | 4.75 | 2.88 | 34.24 | 62.51 | 56.14 | 59.53 | 5.64 | 3.58 |
| Dyn. ($\tau$=0.99) | 24.85 | 79.74 | 42.36 | 68.71 | 5.24 | 3.59 | 36.86 | 82.76 | 73.14 | 78.26 | 5.87 | 4.06 |
| Dyn. (offset=1) | 23.47 | 49.84 | 26.86 | 43.14 | 4.86 | 2.75 | 31.62 | 41.65 | 36.50 | 39.26 | 4.95 | 2.80 |
| Dyn. (offset=2) | **25.26** | 82.55 | 37.88 | 69.36 | 5.13 | 3.47 | **37.09** | 66.52 | 58.98 | 62.99 | 5.60 | 3.99 |

**Table 4:** Performance Comparison of Dynamic $k$ vs. Fixed $k$ Speculative Planning on **OpenAGI** benchmark with **DeepSeek** backbone.

| Mode | Direct-ReACT (Setting 3) | | | | | | CoT-MAD (Setting 4) | | | | | |
|---|---|---|---|---|---|---|---|---|---|---|---|---|
| | $\Delta$T (%) | $\Delta$P (%) | $\Delta$G (%) | $\Delta$Cost (%) | $\overline{MC}$ | $\overline{K}$ | $\Delta$T (%) | $\Delta$P (%) | $\Delta$G (%) | $\Delta$Cost (%) | $\overline{MC}$ | $\overline{K}$ |
| Fix (k=2) | 19.68 | 34.25 | 28.23 | 30.69 | 3.00 | 2.00 | 24.91 | 36.17 | 30.19 | 32.98 | 3.00 | 2.00 |
| Fix (k=4) | 29.19 | 109.41 | 90.92 | 95.15 | 4.92 | 4.00 | 34.22 | 99.61 | 86.99 | 90.01 | 4.94 | 4.00 |
| Fix (k=6) | **30.34** | 171.35 | 134.69 | 142.50 | 6.33 | 6.00 | **36.25** | 144.87 | 128.64 | 128.71 | 6.59 | 6.00 |
| Dyn. ($\tau$=0.5) | 16.88 | 20.54 | 17.16 | **18.74** | 3.89 | 1.69 | 20.03 | 21.90 | 17.58 | **19.99** | 3.78 | 1.54 |
| Dyn. ($\tau$=0.8) | 21.33 | 32.28 | 27.51 | 29.28 | 4.20 | 2.08 | 27.48 | 41.30 | 35.03 | 37.84 | 4.57 | 2.31 |
| Dyn. ($\tau$=0.9) | 23.84 | 47.71 | 40.77 | 42.58 | 4.45 | 2.58 | 30.32 | 52.66 | 45.25 | 48.14 | 4.69 | 2.65 |
| Dyn. ($\tau$=0.95) | 26.25 | 52.32 | 48.18 | 48.13 | 4.78 | 2.83 | 32.87 | 60.91 | 53.43 | 55.88 | 5.12 | 3.12 |
| Dyn. ($\tau$=0.99) | 28.70 | 87.02 | 75.24 | 75.92 | 5.46 | 3.85 | 33.93 | 84.59 | 74.42 | 77.30 | 5.29 | 3.75 |
| Dyn. (offset=1) | 25.07 | 46.30 | 40.92 | 42.40 | 4.39 | 2.58 | 31.79 | 49.08 | 42.90 | 45.41 | 4.48 | 2.80 |
| Dyn. (offset=2) | **29.58** | 78.31 | 68.64 | 69.90 | 4.99 | 3.53 | **34.24** | 76.23 | 66.90 | 69.92 | 5.38 | 3.85 |

**Average Speculative Step ($\overline{K}$)** In addition to peak concurrency ($\overline{MC}$), which captures brief spikes of load, we aim to characterize the *average* concurrency of the planning process over the planning process, as it correlates with sustained system pressure. However, the instantaneous concurrency varies continuously over time. Precisely computing the time-averaged number of concurrent API calls would require fine-grained logs of every call's start and end time, which is often impractical. To address this, we use the average speculative step $\overline{K}$ as a proxy for the true average concurrency. Let $M$ be the total number of planning episodes, and $k_i$ be the speculative steps used in episode $i$. We define

$$\overline{K} = \frac{1}{M} \sum_{i=1}^{M} k_i. \tag{11}$$

Because each target-agent computation is relatively long-running, the $k_i$ speculative calls in episode $i$ tend to overlap in time. In practice, this means the average concurrency during episode $i$ is effectively $k_i$. Averaging these values over all episodes therefore yields a meaningful estimate of the sustained concurrency.

### A.3 ACTUAL PERFORMANCE

Figure 7 visualizes how different speculative planning configurations in Setting 1 of the OpenAGI benchmark impact prompt token increase, generation token increase, and the corresponding reduction in execution time.

*On OpenAGI benchmark*: In **setting 1**, Dyn. (offset = 2) achieves a time reduction comparable to the fastest fixed configuration (Fix ($k = 6$)), while reducing *redundant cost* by **60.65%**. Under **Setting 2**, Dyn. ($\tau = 0.5$) achieves a 23.12% latency reduction while incurring only 19.63% unnecessary cost—saving 34.15% redundant cost compared to the most cost-efficient fixed configuration (Fix, $k = 2$), and delivering better speed. The Dyn. (offset = 2) configuration matches the time savings of the fastest fixed strategy (Fix, $k = 6$), while reducing redundant cost by **53.13%**. Under **setting 3**, Dyn. (offset = 2) delivers time savings comparable to the fastest fixed configuration (Fix, $k = 6$), while reducing redundant cost by **50.95%**. Under **setting 4**, Dyn. (offset = 2) achieves a 97.21% latency

**Table 5:** Performance Comparison of Dynamic $k$ vs. Fixed $k$ Speculative Planning on **Travel Planner** benchmark with **GPT** backbone.

| Mode | Direct-ReACT (Setting 1) | | | | | | CoT-MAD (Setting 2) | | | | | |
|------|------|------|------|------|------|------|------|------|------|------|------|------|
| | $\Delta$T (%) | $\Delta$P (%) | $\Delta$G (%) | $\Delta$Cost (%) | $\overline{MC}$ | $\overline{K}$ | $\Delta$T (%) | $\Delta$P (%) | $\Delta$G (%) | $\Delta$Cost (%) | $\overline{MC}$ | $\overline{K}$ |
| Fix (k=2) | 15.36 | 64.52 | 26.86 | 61.91 | 3.00 | 2.00 | 10.97 | 55.46 | 24.91 | 54.77 | 3.00 | 2.00 |
| Fix (k=4) | 23.48 | 215.76 | 73.93 | 205.98 | 5.00 | 4.00 | **14.53** | 130.46 | 51.66 | 128.93 | 4.62 | 4.00 |
| Fix (k=6) | **24.27** | 364.01 | 109.91 | 346.71 | 6.99 | 6.00 | 12.75 | 159.45 | 55.37 | 157.74 | 5.37 | 6.00 |
| Dyn. ($\tau$=0.5) | 11.47 | 31.96 | 15.28 | **30.74** | 4.28 | 1.46 | 5.61 | 27.46 | 8.24 | **27.02** | 4.09 | 1.30 |
| Dyn. ($\tau$=0.8) | 16.52 | 54.22 | 21.96 | 51.83 | 4.60 | 1.88 | 5.65 | 35.10 | 12.49 | 34.57 | 4.00 | 1.42 |
| Dyn. ($\tau$=0.9) | 19.71 | 72.84 | 29.59 | 69.56 | 4.96 | 2.21 | 9.07 | 55.08 | 22.06 | 54.26 | 4.03 | 1.79 |
| Dyn. ($\tau$=0.95) | 21.04 | 95.01 | 37.10 | 90.72 | 5.19 | 2.55 | 12.74 | 57.54 | 22.35 | 56.65 | 4.00 | 2.10 |
| Dyn. ($\tau$=0.99) | **22.39** | 137.90 | 50.41 | 132.21 | 5.32 | 3.10 | **14.54** | 93.63 | 41.01 | 92.49 | 4.58 | 2.94 |
| Dyn. (offset=1) | 13.55 | 54.95 | 22.22 | 52.13 | 4.31 | 2.30 | 11.86 | 62.65 | 26.30 | 61.96 | 4.08 | 2.16 |
| Dyn. (offset=2) | 16.20 | 89.08 | 30.84 | 84.24 | 4.91 | 3.22 | 12.75 | 105.28 | 43.65 | 104.16 | 4.33 | 2.98 |

reduction relative to the fastest fixed baseline (Fix $k = 6$), while concurrently reducing redundant cost by approximately 45.68%.

*On Travel Planner benchmark*: In **setting 1**, Dyn. ($\tau = 0.8$) achieves slightly faster time reduction than Fix ($k = 2$), while reducing redundant cost by 16.28%. Under **setting 2**, Dyn. ($\tau = 0.99$) achieves similar latency compared to both Fix ($k = 4$) and Fix ($k = 2$), while separately reducing redundant cost by 28.26% and 41.37% respectively.

Experiment results also demonstrate that DSP achieves significantly more efficient concurrency utilization compared to fixed $k$ baselines. DSP achieves 40% lower sustained system pressure($\overline{K}$) in its fastest configurations (Dyn. ($\tau = 0.99$), Dyn. (offset = 2)), compared to the fastest fixed-$k$ setting (Fix ($k = 6$)), while maintaining comparable latency reduction. This demonstrates that dynamic policies can minimize persistent system load without sacrificing acceleration performance. All dynamic variants maintain $\overline{MC}$ below 6, outperforming the fastest Fix ($k = 6$) configuration.

### A.4 BIASED $k$ PREDICTION $v.s.$ $k$ WITH BIASED OFFSET

**Tunable trade-off control** The $\tau$ hyperparameter provides fine-grained control over the latency-cost trade-off by enabling three operational modes, defining a smooth continuum of operating points, forming a Pareto frontier between latency and token cost. This flexibility allows practitioners to adjust system behavior dynamically to meet varying operational requirements and economic constraints.

- **High-Performance Mode** ($\tau = 0.99$, $\tau = 0.95$): It provides over 90% of the time savings compared to fastest setting (even exceeding its performance) while saving 30% total cost.
- **Balanced Mode** ($\tau = 0.9$, $\tau = 0.8$): It achieves over 80% of latency reduction, while on average halves total cost compared to the fastest setting.
- **Economy Mode** ($\tau = 0.5$): It maintains non-trivial acceleration while reducing cost overhead compared to the most cost-efficient fixed setting.

All three modes are supported within a unified DSP framework by simply tuning $\tau$, without altering the algorithm. This tunability allows DSP to meet different application requirements. In DSP, $\tau$ provides a smooth continuum of operating points, effectively generating a Pareto curve of latency vs. token cost from which one can choose according to preferences. As an interpretable hyperparameter, $\tau$ offers clear deployment guidance: higher values promote larger $k$ predictions, while lower values encourage conservative steps, as detailed in Section 4.3. As shown in Table 3, Table 4, and Table 5, the average speculation step $\overline{K}$ increases with $\tau$, confirming its practical and theoretical role in governing speculative planning behavior.

**Biased $k$ Prediction v.s. $k$ with Biased Offset** We examines two distinct approaches for controlling $k$: training-time bias adjustment through expectile regression and inference-time direct offset application. Both methods enable practitioners to navigate the latency-cost tradeoff, though with different optimization characteristics.

*Applying a fixed positive offset to the unbiased predicted value of $k$ offers a computationally efficient and conceptually straightforward approach to balancing acceleration and cost efficiency.* In OpenAGI benchmark, DSP(offset=1) achieves a latency reduction comparable to DSP($\tau = 0.95$). Similarly,

**Table 6:** Warm-up Phase Accuracy (First 13 Tasks).

**(a)** OpenAGI

| Config | Task 1 | Task 2 | Task 3 | Task 4 | Task 5 | Task 6 | Task 7 | Task 8 | Task 9 | Task 10 | Task 11 | Task 12 | Task 13 |
|---|---|---|---|---|---|---|---|---|---|---|---|---|---|
| Direct-ReAct (GPT-4.1-mini) | 0 | 0.42 | 0.67 | 0.67 | 1 | 0.75 | 0.6 | 1 | 1 | 1 | 0.8 | 0.5 | 0.83 |
| Direct-ReAct (DeepSeek) | 0 | 0 | 0 | 0 | 0 | 0.33 | 0.67 | 0 | 0.5 | 0.33 | 0.67 | 0.5 | 0.67 |
| CoT-MAD (GPT-4.1-mini) | 0 | 0 | 0.5 | 0.5 | 0.67 | 0.25 | 0.38 | 0.75 | 0.5 | 0.8 | 0.57 | 0.6 | 0.8 |
| CoT-MAD (DeepSeek) | 0 | 0 | 0 | 0 | 0 | 0 | 0 | 0 | 0.33 | 0.33 | 0.25 | 0.33 | 1 |

**(b)** TravelPlanner

| Config | Task 1 | Task 2 | Task 3 | Task 4 | Task 5 | Task 6 | Task 7 | Task 8 | Task 9 | Task 10 | Task 11 | Task 12 | Task 13 |
|---|---|---|---|---|---|---|---|---|---|---|---|---|---|
| Direct-ReAct (GPT-4.1-mini) | 0 | 0 | 0.4 | 0.38 | 0.44 | 0.88 | 1 | 0.57 | 0.75 | 0.62 | 0.67 | 0.57 | 0.86 |
| CoT-MAD (GPT-4.1-mini) | 0 | 0 | 0.62 | 1 | 0.29 | 1 | 0.43 | 0.83 | 0.57 | 0.71 | 0.71 | 1 | 0.57 |

DSP(offset=2) achieved 97% acceleration by the optimal fixed-$k$ setting where $k = 6$, while saving up to 28% cost. These results demonstrate that simple post-prediction adjustment yields robust performance improvements across both latency and cost metrics.

*However, the principal advantage of biased-$k$ prediction implemented through training-time expectile regression lies in its capacity for more granular control over the efficiency-cost continuum.* Experimental results indicate that moderate training parameters ($\tau \in 0.8, 0.9$) delivers acceleration profiles that cannot be precisely matched by integer offset adjustments. These intermediate settings enable more nuanced optimization configurations that achieve reasonable latency reduction with substantially lower operational costs than their offset-based counterparts (offset $\in +1, +2$).

The choice between these approaches ultimately depends on the specific operational requirements and constraints. Training-time bias adjustment provides finer-grained control and potentially better optimization at intermediate points on the efficiency-cost spectrum, particularly valuable for applications with precise performance requirements. Conversely, inference-time offset application offers implementation simplicity, immediate adjustability, and strong performance at discrete operating points, making it suitable for deployments where straightforward configuration and rapid adaptation are prioritized.

## A.5    ONLINE WARMUP

In our experiment, we begin with a randomly initialized DistilBERT model at the initial stage of every setting and its predictions are naturally zero at the start. This results in the entire pipeline executing the target agent in a sequential manner during warm-up tasks without any additional cost increase. To expedite the reduction of warmup time, we start collecting training episodes online as the pipeline runs. Once enough episodes are gathered to meet the predefined batch size, the model undergoes its first training update.

Our analysis across different settings reveals that the warmup duration varies depending on the API backbone. As shown in Table 6, when using GPT-4.1-mini, it typically requires around 3–5 tasks to warm up, as GPT-4.1-mini tends to generate longer sequences to complete each task. It enables faster collection of a sufficient number of training episodes. In contrast, when using DeepSeek-series models, the warmup phase is slower, requiring approximately 8–10 tasks to collect enough training data. This discrepancy highlights the differences in the response times of different APIs and the resulting impact on the warmup efficiency.

## A.6    PREDICTOR ACCURACY OVER ONLINE TRAINING

To assess the effectiveness and stability of our online learning procedure, we examine predictor accuracy on the OpenAGI benchmark across sequential task segments. Table 7 summarizes the mean and standard deviation of accuracy within non-overlapping 50-task windows, and Figure 5 visualizes the global trend with a smoothed trajectory.

In the early stage (tasks 1–100), the predictor exhibits moderate accuracy and relatively high variance, reflecting the initial adaptation to unseen task distributions. As training progresses, accuracy consistently improves across all settings, accompanied by a steady reduction in variance. By the later

**Table 7:** Predictor Accuracy per 50-Task Segment on the OpenAGI benchmark (Mean $\pm$ SD).

| Task | Direct-ReAct (GPT) | Direct-ReAct (DeepSeek) | CoT-MAD (GPT) | CoT-MAD (DeepSeek) |
|---|---|---|---|---|
| 1–50 | 0.600±0.351 | 0.465±0.317 | 0.641±0.342 | 0.459±0.334 |
| 51–100 | 0.477±0.363 | 0.499±0.331 | 0.693±0.349 | 0.501±0.346 |
| 101–150 | 0.689±0.256 | 0.701±0.352 | 0.675±0.285 | 0.564±0.334 |
| 151–200 | 0.693±0.189 | 0.593±0.326 | 0.575±0.217 | 0.570±0.329 |
| 201–250 | 0.755±0.207 | 0.657±0.280 | 0.621±0.198 | **0.704±0.239** |
| 251–300 | 0.724±0.169 | **0.762±0.207** | 0.693±0.135 | 0.674±0.237 |
| 301–312 | **0.761±0.205** | 0.707±0.201 | **0.760±0.130** | 0.698±0.259 |

**(a)** Direct-ReAct (GPT-4.1-mini)

**(b)** Direct-ReAct (DeepSeek)

**(c)** CoT-MAD (GPT-4.1-mini)

**(d)** CoT-MAD (DeepSeek)

**Figure 5:** Predictor accuracy trends over online training across different settings. In all cases (Direct-ReAct and CoT-MAD with GPT-4.1-mini or DeepSeek), the predictor exhibits a clear upward trajectory in accuracy alongside decreasing variance, indicating improved generalization to unseen tasks and stable convergence without late-stage drift.

segments (tasks 200–300), the predictor achieves substantially higher mean accuracy (0.70–0.76) while maintaining much lower variance, indicating convergence and stability.

These results demonstrate that online updates reliably enhance the predictor's generalization ability to unseen tasks. Importantly, we observe no evidence of performance drift or instability in later stages. Instead, the predictor becomes increasingly accurate and stable, ensuring dependable speculative planning across diverse benchmarks.

## A.7 EVALUATION ON NON-CONTEXTUAL DYNAMIC HEURISTICS

To further demonstrate the advantage of our **Dynamic Speculative Planning (DSP)**, we implemented a non-contextual reward-driven black-box optimization heuristic using Bayesian Optimization (BO). Unlike DSP, this method does not exploit contextual state features, but instead attempts to learn a mapping from past speculative outcomes to effective $k$-values. Specifically, BO maintains a surrogate model (Random Forest Regressor) to predict the expected reward for different $k$-values based on past selection outcomes. The reward is defined as:

$$R(k) = \frac{1}{|k - k^*| + 1},$$

where $k^*$ denotes the ground-truth speculative depth. This formulation incentivizes the selection of $k$ values closer to the true horizon. For the exploration–exploitation tradeoff, we adopted a lightweight $\epsilon$-Greedy acquisition strategy ($\epsilon = 0.1$), which maintains efficiency during online updates.

**Table 8:** Performance of Bayesian Optimization and DSP on OpenAGI (First 200 Tasks).

| Method | Direct-ReAct (GPT-4.1-mini) | | | | Direct-ReAct (DeepSeek) | | | | CoT-MAD (GPT-4.1-mini) | | | |
|---|---|---|---|---|---|---|---|---|---|---|---|---|
| | Time | Prompt | Gen | Cost | Time | Prompt | Gen | Cost | Time | Prompt | Gen | Cost |
| **Fixed-k** | | | | | | | | | | | | |
| $k = 2$ | 1.00 | 1.00 | 1.00 | 1.00 | 1.00 | 1.00 | 1.00 | 1.00 | 1.00 | 1.00 | 1.00 | 1.00 |
| $k = 4$ | 0.90 | 1.47 | 1.21 | 1.35 | 0.85 | 1.49 | 1.44 | 1.46 | 0.81 | 1.35 | 1.34 | 1.32 |
| $k = 6$ | 0.88 | 2.00 | 1.47 | 1.74 | 0.84 | 1.85 | 1.76 | 1.82 | 0.77 | 1.69 | 1.63 | 1.62 |
| **DSP** | | | | | | | | | | | | |
| $\tau = 0.5$ | **0.99** | 0.94 | 0.96 | **0.94** | **1.02** | 0.95 | 0.96 | **0.96** | **0.92** | 1.03 | 1.02 | **1.03** |
| $\tau = 0.8$ | 0.94 | 1.06 | 1.03 | 1.04 | 0.98 | 1.02 | 1.04 | 1.02 | 0.84 | 1.10 | 1.10 | 1.10 |
| $\tau = 0.9$ | 0.93 | 1.14 | 1.06 | 1.10 | 0.92 | 1.09 | 1.08 | 1.09 | 0.82 | 1.15 | 1.14 | 1.14 |
| $\tau = 0.95$ | 0.91 | 1.22 | 1.13 | 1.17 | 0.91 | 1.20 | 1.22 | 1.21 | 0.81 | 1.30 | 1.29 | 1.28 |
| $\tau = 0.99$ | 0.90 | 1.37 | 1.21 | 1.29 | 0.87 | 1.42 | 1.41 | 1.41 | 0.81 | 1.39 | 1.37 | 1.36 |
| offset=1 | 0.91 | 1.15 | 1.08 | 1.11 | 0.93 | 1.12 | 1.14 | 1.13 | 0.84 | 1.15 | 1.13 | 1.13 |
| offset=2 | 0.90 | 1.35 | 1.17 | 1.27 | 0.85 | 1.30 | 1.31 | 1.30 | 0.76 | 1.33 | 1.30 | 1.30 |
| **Heuristic** | | | | | | | | | | | | |
| BO | **1.03** | 0.98 | 0.98 | **0.97** | **1.09** | 0.98 | 0.94 | **0.95** | **1.10** | 1.03 | 0.95 | **1.00** |

As shown in Table 8, in Direct-ReAct setting with GPT backbone, BO achieves performance comparable to the fixed $k = 2$ baseline, but fails to realize the dynamic acceleration potential of DSP. Compared to our DSP variant with $\tau = 0.5$, BO yields a similar latency reduction yet incurs **10% higher total cost**. Conversely, when compared with DSP at $\tau = 0.8$, BO matches token cost but delivers **significantly less acceleration**. Direct-ReAct setting with DeepSeek backbone and CoT-MAD setting with GPT backbone only contains first 250 tasks. In these two settings, BO consistently lags behind DSP variants as it delivers substantially worse acceleration.

These results demonstrate that non-contextual methods like BO cannot effectively capture the nuanced state information from dual agents, which is critical for accurate speculation prediction. This limitation underscores the necessity of employing lightweight neural networks such as DistilBERT to encode contextual features and learn complex state-dependent patterns that simple black-box optimization approaches fail to exploit. DSP maintains a superior Pareto frontier, achieving a better balance between efficiency and resource usage through this context-aware prediction capability.

## A.8 SUPERVISED VS. REINFORCEMENT LEARNING FOR SPECULATIVE STEP PREDICTION

We investigate two alternative formulations for speculative step prediction: supervised learning (SFT) and reinforcement learning (RL) with TD-based updates. While both are supported within our framework, we ultimately adopt the RL formulation for two key reasons.

Theoretically, supervised learning arises as a special case of our framework. When $\lambda = 1$ and $\gamma = 1$, the TD target reduces to the rollout length $G_t = k$, thus TD-learning reduces to online supervised learning. Additionally, $\lambda$-returns provide a tunable bias–variance tradeoff between Monte Carlo and bootstrapped estimates, offering greater flexibility than pure regression. Overall, our learning framework is a more generalized framework.

Emprically, training with fully supervised targets (i.e., $\lambda = 1, \gamma = 1$) produced overly conservative predictors. As shown in Table 9, while the Dyn. (SFT) model achieves similar or even lower token cost savings as our TD-trained variant (Dyn. ($\tau$=0.5)), it consistently underperforms in terms of latency reduction. In most cases, the performance was even slower than a fixed $k = 2$ baseline, which is the minimal non-trivial setting. Additionally, the Dyn. (SFT) model achieves lower accuracy across all settings compared to Dyn. ($\tau$=0.5) The experimental findings reveal the limitations of the supervised learning approach in this context. We attribute this to high variance in rollout-derived $k$ labels and the averaging tendency of regression, which causes SFT models to underestimate $k$ in uncertain regions.

**Table 9:** Performance comparison of SFT vs. RL strategies on the **OpenAGI** benchmark.

| Backbone | Mode | Direct-ReAct | | | | CoT-MAD | | | |
|---|---|---|---|---|---|---|---|---|---|
| | | T(×) | P(×) | G(×) | Cost(×) | T(×) | P(×) | G(×) | Cost(×) |
| GPT-4.1-mini | Fix ($k$=2) | 1.00 | 1.00 | 1.00 | 1.00 | 1.00 | 1.00 | 1.00 | 1.00 |
| | Dyn. ($\tau$=0.5) | 1.01 | 0.87 | 0.91 | **0.88** | 0.98 | 0.95 | 0.95 | **0.95** |
| | Dyn. (SFT) | 1.05 | 0.86 | 0.91 | **0.88** | 1.02 | 0.96 | 0.96 | 0.96 |
| DeepSeek | Fix ($k$=2) | 1.00 | 1.00 | 1.00 | 1.00 | 1.00 | 1.00 | 1.00 | 1.00 |
| | Dyn. ($\tau$=0.5) | 1.04 | 0.93 | 0.94 | **0.93** | 1.07 | 0.94 | 0.94 | **0.94** |
| | Dyn. (SFT) | 1.09 | 0.96 | 0.95 | 0.96 | 1.11 | 1.01 | 1.00 | 1.00 |

## A.9 DSP COST BROKEN-DOWN

The cost breakdown data for Fix and Dynamic Speculative Planning across different asymmetry parameters $\tau$ and different offset is presented in Table.10. In both DSP settings, the cost structure maintains the same general pattern observed in fixed-$k$ Speculative Planning, with the majority of tokens being consumed by target agent operations in input prompt; but $\tau$ parameter effectively controls the aggressiveness of speculation, showing clear patterns in cost reduction:

At the most conservative setting where $\tau = 0.5$, the DSP reduces the average redundant token consumption from +10,748.3 to +672.2 in setting 1 and from +20,424.0 to +8,508.7 redundant tokens in setting 2, respectively, comparing with the most conservative fixed-$k$ setting where $k = 2$. *This represents an around 71% reduction in redundant token consumption.* At the most aggressive setting where $\tau = 0.99$, the DSP also reduces the average total token consumption from +52,664.6 to +14,452.6 in setting 1 and from +96,927.3 to +50,977.8 in setting 2, respectively, comparing with the most aggressive fixed-$k$ setting where $k = 6$. *This also represents an around 50% reduction in redundant token consumption.* The graduated scaling of costs with the $\tau$ parameter provides practitioners with fine-grained control over the efficiency-cost balance, enabling precise system calibration that fixed-$k$ approaches cannot match.

## A.10 LIMITATIONS

While our Dynamic Speculative Planning framework demonstrates significant improvements in balancing acceleration and cost efficiency, several important limitations warrant acknowledgment and suggest directions for future research. First, our current implementation does not adaptively select or optimize the approximation agent itself. The speed, accuracy, and availability characteristics of this agent represent crucial factors in overall system performance. Different approximation agents may be better suited for specific tasks or planning stages, and dynamically selecting from a pool of candidate agents could further enhance both acceleration and cost-efficiency. Future work could explore meta-learning approaches that automatically select or even generate task-specific approximation agents based on observed performance patterns.

Second, our current rejection criteria for verifying speculation steps remain relatively strict, potentially limiting achievable acceleration. Agent planning often admits multiple valid solution paths to accomplish the same task, yet our verification mechanism may incorrectly reject valid alternative approaches proposed by the approximation agent when they differ syntactically from the target agent's preferred approach. This high false negative rate in the rejection criteria unnecessarily constrains parallelization opportunities. Future research could explore semantic verification mechanisms that recognize functionally equivalent actions despite surface-level differences, or could implement more sophisticated verification strategies that consider the downstream effects of actions rather than just their immediate representation.

Finally, our evaluation focused primarily on two standard agent benchmarks; more extensive testing across diverse application domains would provide greater confidence in the generalizability of our findings.

**Table 10:** Speculative Planning cost broken-down on **OpenAGI** benchmark with **GPT** backbone.

**(a)** Direct-ReAct (Setting 1)

| Plan Type | Approx Prompt | Approx Generation | Target Prompt | Target Generation | Total Prompt | Total Generation | Total Token |
|---|---|---|---|---|---|---|---|
| Normal Plan | 6040.3 | 108.6 | 12715.6 | 1906.7 | 18755.9 | 2015.3 | 20771.2 |
| Fix (k=2) | 10759.8 | 165.7 | 18162.7 | 2431.2 | 28922.5 | 2596.9 | 31519.4 |
| Fix (k=4) | 19894.5 | 281.9 | 26610.1 | 2963.1 | 46504.6 | 3245.1 | 49749.7 |
| Fix (k=6) | 34124.0 | 369.2 | 35079.7 | 3862.9 | 69203.7 | 4232.1 | 73435.8 |
| Dyn. ($\tau = 0.5$) | 6457.7 | 115.0 | 13022.8 | 1965.6 | 19480.5 | 2081.6 | 21443.4 |
| Dyn. ($\tau = 0.8$) | 7454.8 | 130.9 | 14348.4 | 1994.6 | 21803.1 | 2125.5 | 23928.6 |
| Dyn. ($\tau = 0.9$) | 8745.6 | 148.9 | 16062.6 | 2185.7 | 24808.2 | 2334.6 | 27142.8 |
| Dyn. ($\tau = 0.95$) | 9226.2 | 154.7 | 16407.2 | 2176.9 | 25633.4 | 2331.6 | 27965.0 |
| Dyn. ($\tau = 0.99$) | 12307.0 | 195.1 | 20140.0 | 2581.7 | 32447.0 | 2776.8 | 35223.8 |
| Dyn. (offset = 1) | 9576.3 | 163.9 | 17381.4 | 2293.0 | 26957.7 | 2456.9 | 29414.7 |
| Dyn. (offset = 2) | 12999.8 | 206.1 | 20254.3 | 2457.7 | 33254.1 | 2663.8 | 35917.9 |
| **Redundant Cost** | | | | | | | |
| $\Delta$ Fix (k = 2) | +4719.5 | +57.1 | +5447.1 | +524.5 | +10166.6 | +581.6 | +10748.3 |
| $\Delta$ Fix (k = 4) | +13854.2 | +173.3 | +13894.4 | +1056.5 | +27748.7 | +1229.8 | +28978.5 |
| $\Delta$ Fix (k = 6) | +28083.7 | +260.6 | +22364.1 | +1956.2 | +50447.8 | +2216.8 | +52664.6 |
| $\Delta$ **Dyn. ($\tau = 0.5$)** | +417.4 | +6.4 | +307.2 | +58.9 | **+724.6** | **+66.3** | **+672.2** |
| $\Delta$ Dyn. ($\tau = 0.8$) | +1414.5 | +22.3 | +1632.8 | +87.9 | +3047.2 | +110.2 | +3157.4 |
| $\Delta$ Dyn. ($\tau = 0.9$) | +2705.3 | +40.3 | +3347.0 | +279.0 | +6052.3 | +319.3 | +6371.6 |
| $\Delta$ Dyn. ($\tau = 0.95$) | +3185.9 | +46.1 | +3691.6 | +270.2 | +6877.5 | +316.3 | +7193.8 |
| $\Delta$ Dyn. ($\tau = 0.99$) | +6266.7 | +86.5 | +7424.4 | +675.0 | +13691.1 | +761.5 | +14452.6 |
| $\Delta$ Dyn. (offset = 1) | +3536.0 | +55.3 | +4665.8 | +386.3 | +8201.8 | +441.6 | +8643.5 |
| $\Delta$ Dyn. (offset = 2) | +6959.5 | +97.5 | +7538.7 | +551.0 | +14498.2 | +648.5 | +15146.7 |

**(b)** CoT-MAD (Setting 2)

| Plan Type | Approx Prompt | Approx Generation | Target Prompt | Target Generation | Total Prompt | Total Generation | Total Token |
|---|---|---|---|---|---|---|---|
| Normal Plan | 5422.5 | 1701.3 | 35968.4 | 7332.8 | 41390.9 | 9034.0 | 50424.9 |
| Fix (k=2) | 8158.9 | 2435.4 | 50373.5 | 9881.1 | 58532.4 | 12316.5 | 70848.9 |
| Fix (k=4) | 13799.2 | 3884.8 | 70283.3 | 13253.9 | 84082.5 | 17138.7 | 101221.2 |
| Fix (k=6) | 26065.3 | 5466.1 | 97668.0 | 18152.8 | 123733.3 | 23618.9 | 147352.2 |
| Dyn. ($\tau = 0.5$) | 6429.6 | 1946.9 | 42200.3 | 8356.7 | 48629.9 | 10303.7 | 58933.6 |
| Dyn. ($\tau = 0.8$) | 7671.9 | 2330.3 | 48703.2 | 9577.7 | 56375.1 | 11908.0 | 68283.1 |
| Dyn. ($\tau = 0.9$) | 8682.5 | 2590.9 | 52586.8 | 10255.2 | 61269.3 | 12846.1 | 74115.5 |
| Dyn. ($\tau = 0.95$) | 10417.5 | 3043.1 | 58605.2 | 11353.2 | 69022.7 | 14396.4 | 83419.1 |
| Dyn. ($\tau = 0.99$) | 13857.5 | 3848.7 | 70281.1 | 13415.5 | 84138.5 | 17264.2 | 101402.7 |
| Dyn. (offset = 1) | 8978.2 | 2699.9 | 54749.3 | 10715.0 | 63727.4 | 13414.9 | 77142.3 |
| Dyn. (offset = 2) | 11256.8 | 3276.3 | 61864.3 | 11828.1 | 73121.1 | 15104.4 | 88225.6 |
| **Redundant Cost** | | | | | | | |
| $\Delta$ Fix (k = 2) | +2736.4 | +734.1 | +14405.1 | +2548.3 | +17141.5 | +3282.4 | +20424.0 |
| $\Delta$ Fix (k = 4) | +8376.7 | +2183.5 | +34314.9 | +5921.2 | +42691.6 | +8104.7 | +50796.3 |
| $\Delta$ Fix (k = 6) | +20642.8 | +3764.9 | +61699.6 | +10820.0 | +82342.4 | +14584.9 | +96927.3 |
| $\Delta$ **Dyn. ($\tau = 0.5$)** | +1007.1 | +245.6 | +6231.9 | +1023.9 | **+7239.0** | **+1269.7** | **+8508.7** |
| $\Delta$ Dyn. ($\tau = 0.8$) | +2249.4 | +629.0 | +12734.8 | +2244.9 | +14984.2 | +2874.0 | +17858.2 |
| $\Delta$ Dyn. ($\tau = 0.9$) | +3260.0 | +889.6 | +16618.4 | +2922.4 | +19878.4 | +3812.1 | +23690.6 |
| $\Delta$ Dyn. ($\tau = 0.95$) | +4995.0 | +1341.8 | +22636.8 | +4020.4 | +27631.8 | +5362.4 | +32994.2 |
| $\Delta$ Dyn. ($\tau = 0.99$) | +8435.0 | +2147.4 | +34312.7 | +6082.7 | +42747.6 | +8230.2 | +50977.8 |
| $\Delta$ Dyn. (offset = 1) | +3555.7 | +998.6 | +18780.9 | +3382.2 | +22336.5 | +4380.9 | +26717.4 |
| $\Delta$ Dyn. (offset = 2) | +5834.3 | +1575.0 | +25895.9 | +4495.3 | +31730.2 | +6070.4 | +37800.7 |

Despite these limitations, our work represents a significant step toward making accelerated LLM-based agents more practical for real-world deployment by addressing the critical tension between latency reduction and cost efficiency while providing unprecedented user control over system behavior.

### A.11 BROADER IMPACT

Dynamic Speculative Planning contributes to the broader effort of making AI agents more practical and accessible for real-time applications. By substantially reducing response latency while providing fine-grained control over operational costs, our approach can help the deployment of LLM-based agents in time-sensitive contexts. For examples: educational applications could benefit from more responsive tutoring agents that provide immediate feedback while managing computational expenses within institutional constraints; healthcare systems might leverage faster diagnostic support agents without compromising accuracy or incurring prohibitive costs; customer service platforms could deploy assistants with response times approaching human conversation pace.

### A.12 WALL CLOCK TIME ANALYSIS

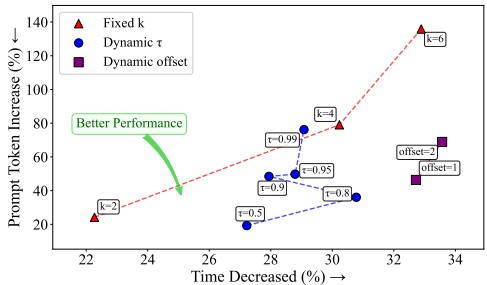 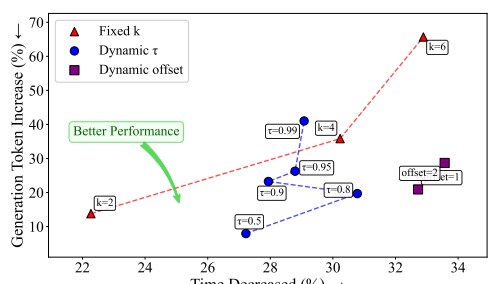

**(a)** Prompt Token Increase vs Time Decrease            **(b)** Generation Token Increase vs Time Decrease

**Figure 6:** Performance trade-off between acceleration and token increase between Fix-$k$, Dynamic-$\tau$ and Dynamic-offset SP **within first 50 tasks**. (a) Prompt token and (b) Generation token increase vs. latency reduction, both relative to a normal-plan baseline.

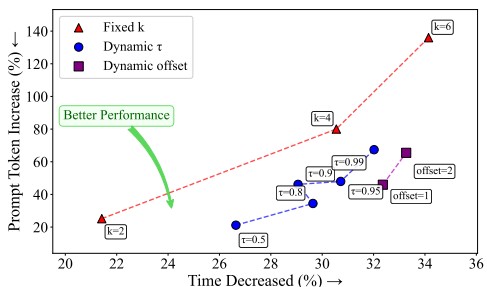 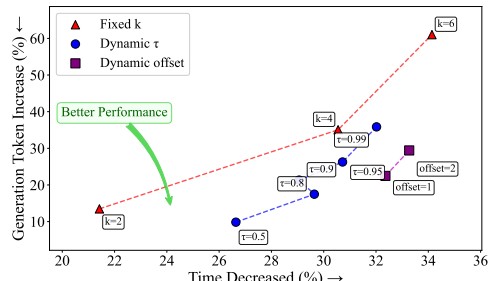

**(a)** Prompt Token Increase vs Time Decrease            **(b)** Generation Token Increase vs Time Decrease

**Figure 7:** Performance trade-off between acceleration and token increase between Fix-$k$, Dynamic-$\tau$ and Dynamic-offset SP **within first 100 tasks**. (a) Prompt token and (b) Generation token increase vs. latency reduction, both relative to a normal-plan baseline.

We provide a performance comparison between DSP and fixed-k baselines on the first 50 and first 100 tasks.

Within the first 50 tasks, all DSP configurations except $\tau = 0.99$ already reach the Pareto frontier, and within the first 100 tasks, all DSP variants lie on the frontier.

These experiments results demonstrate that DSP quickly stabilizes near the optimal latency-cost region, achieving Pareto-efficient behavior early in training without adding wall-clock cost.

