# OpenReview forum: "Dynamic Speculative Agent Planning"
_ICLR.cc/2026/Conference — ICLR 2026 Poster_

### Official Review · Reviewer_J1uY · 2025-10-28

**Soundness:** 3
**Presentation:** 3
**Contribution:** 3
**Rating:** 4
**Confidence:** 3

**Summary:**

This paper proposes a framework to accelerate large language model (LLM)-based agents by reducing inference latency and cost—two major bottlenecks in real-world deployment. The authors build on speculative execution (a technique where a lightweight model “guesses” future steps while a stronger model verifies them) and extend it to multi-step agent planning. Existing speculative planning methods use fixed speculation steps (k), which can either waste resources (if k is too high) or fail to accelerate enough (if k is too low). The method proposed here, DSP, introduces a dynamic, reinforcement-learning-based mechanism that adaptively chooses the optimal speculation step size during runtime. However, it still needs broader empirical validation, deeper real-world integration, and more nuanced cost modeling to fully realize its impact.

**Strengths:**

1. The use of online reinforcement learning (Temporal Difference (TD) learning with λ-returns) to dynamically adjust speculative step size (k) is an elegant solution to a well-known inefficiency in speculative planning. The method’s real-time adaptability—learning directly during deployment rather than via costly offline training—is both practical and technically forward-looking.
2. The proposed method DSP does not require router training, parallelization setup, or prompt engineering, making it directly applicable to many existing agent frameworks.

**Weaknesses:**

1. The evaluation is confined to OpenAGI and TravelPlanner, which are still benchmarks, not real-life systems. Generalization to highly dynamic or human-interactive domains (e.g., robotics) remains untested.
2. The paper defines a simple reward function (reward = 1 per correct speculative step), which might oversimplify complex planning dynamics. Real-world task rewards often involve long-term dependencies or delayed outcomes, which may challenge the proposed TD formulation.
3. The experiments assume fixed per-token costs, but API pricing, context-length, and throughput pricing in practice vary nonlinearly. The claim of “30% cost reduction” could fluctuate under real billing conditions.
4. The baselines focus on fixed speculative methods. A stronger evaluation might compare DSP with distillation-based acceleration or dual process models (System-1.x) for a more comprehensive view.

**Questions:**

1. The reward function sets R=1 for correct steps and 0 otherwise, which seems coarse. Have you explored more informative reward signals (e.g., scaled by prediction confidence or cost impact)?
2. How do you ensure stability when the predictor is updated asynchronously while being used for inference?
3. When the predicted step k is far from optimal, how do you mitigate oscillations or compounding inefficiency (especially early in training)?
4. While DSP aims to reduce token cost, it adds overhead for training and maintaining the predictor. How significant are these infrastructure costs in large-scale deployments?
5. The paper provides code, but could you report how sensitive the results are to λ, learning rate, and batch size?

---

> ### Author Response · Authors · 2025-11-20
> **Response for weakness 1~4 & question 1**
>
> We sincerely appreciate your thoughtful review and constructive feedback. We are especially grateful that you highlighted the strengths of our work, including **the elegant and novel online RL framework, real-time adaptability during deployment, and the practical plug-and-play design directly applicable to existing agent frameworks**.
>
> ---
> > **W #1. Highly Dynamic Benchmarks (eg. robotics)**
>
> We appreciate the reviewer’s question. We would also like to clarify that highly dynamic domains such as robotics tasks often involve irreversible or high-stakes actions, where speculative execution must be applied with greater caution.
>
> However, to **deploy in real life scenarios**, it would require more reliable semantic verification, safety-aware rollback mechanisms, or sophisticated world simulators to prevent false speculative actions from causing irreversible environmental changes or task-level failures, **which the current technical contribution does not focus on and do not contain**.
>
> We view this as an **important and promising direction for future work**, aiming to adapt our dynamic speculative planning framework for safe deployment in interactive and real-world settings.
>
> ---
> > **W #2.& Q #1. Reward Signal Design**
>
> We thank the reviewers for this insightful feedback. Our reward formulation is **simple, dense, interpretable and directly aligned with the objective of maximizing correct speculative steps**.
>
> In RL, the reward function provides immediate, state-action-dependent feedback, while the return captures long-term cumulative outcomes. The reward function is a function of state-action pairs (i.e., $R(s,a)$), thus it is orthogonal to the environment dynamics.
>
> Our reward design ($R(s,a) = 1$ for correct speculative steps, $0$ otherwise) is **intentionally simple, dense, and rule-based to ensure that the return $G_t$ directly measures the number of successful speculative steps, maintaining interpretability and alignment with the objective**. This simplicity avoids overfitting to heuristic signals and has proven effective in practice.
>
> While more sophisticated reward shaping (e.g., confidence-weighted or cost-aware rewards) could be explored in future work, our current formulation strikes a practical balance between efficacy and robustness.
>
> ---
> > **W #3 Token & Pricing Cost**
>
> We thank the reviewer for raising this important concern. **Our evaluation does include API and throughput pricing** (As discussed in Section 5.1 and detailed in Appendix A.2–A.3).
>
> Our metrics include T(×), P(×), G(×), and Cost(×) (mentioned in lines 396-397), representing relative latency, prompt token usage, generation token usage, and **monetary cost**, respectively. The monetary cost metric explicitly **combines token usage (both prompt and generation tokens) with actual API pricing** to reflect throughput pricing under real billing conditions.
>
> The detailed definitions and calculation formulas for these metrics are provided in Appendix A.2, where we specify how API pricing is incorporated into our cost calculations. Our main experimental results (Section 5.1) and actual performance evaluation (Appendix A.3) provide comprehensive comparisons across all these metrics.
>
>
> ---
> > **W #4. System 1.x and EcoAct baselines**
>
>
> System-1.x and EcoAct address related efficiency challenges, but **through fundamentally different approaches that require either extensive offline training or intrusive prompt modifications**, whereas DSP achieves adaptive efficiency in a **fully model-agnostic and plug-and-play fashion**.
>
> **(1) System-1.x**: relies on an offline data generation stage that constructs training datasets for three components: a controller, a System-1 planner, and a System-2 planner. This design comes with **significant offline data-collection cost and fine-tuning overhead, and its generalization is bounded by the availability of such search traces.**
>
> In contrast, **DSP performs fully online adaptation on top of pretrained agents without requiring any costly offline data collection or component re-training**. It can be applied to new agent configurations or LLM backbones in a plug-and-play manner, since all learning happens dynamically during task execution rather than through domain-specific fine-tuning.
>
> **(2) EcoAct: requires modifying the agent’s internal prompt and action schema**, by adding a meta-tool (“tool register”) that modifies the agent’s tool-selection process. Moreover, while it reduces context length, this **compression is not lossless**, as it may drop tool descriptions or contextual information that can **affect reasoning fidelity**.

---

> ### Author Response · Authors · 2025-11-20
> **Response for question 2~4**
>
> > **Q #2. How to ensure stability during asynchronous update**
>
> **(1) First, training typically completes before the next prediction is needed and we implement proper synchronization mechanisms**.
>
> Each training thread takes only approximately **1.5 seconds** which is shorter than a single target agent inference step (GPT-based agents typically take 3 seconds per step; DeepSeek-based agents take over 20 seconds per step) and **significantly shorter than one complete speculative episode** (which usually spans several agent steps).
>
> **This means training typically completes before the next prediction is needed and the latest weights have been atomically synchronized with the inference model**. The asynchronous multi-threaded mechanism is designed primarily to guarantee that predictor operations never block the main agent planning pipeline to maximize acceleration.
>
> **(2) Second, we apply the replay buffer which is well known to improve stability[1]**.
>
> **(3) Moreover, the empirical evaluation further confirms stability**. Predictor accuracy serves as a direct measure of training stability. As shown in Appendix A.6, **predictor accuracy gradually converges around 120 tasks, with mean accuracy increasing and variance decreasing** as more data accumulates. This demonstrates that our asynchronous training mechanism does not introduce instability or oscillations in practice.
>
> [1] "Human-level control through deep reinforcement learning." nature 518.7540 (2015): 529-533.
>
>
> ---
> > **Q #3. Effect of Prediction Error**
>
> **(1) Our predictor does not exhibit oscillation behavior during the warm-up phase**.
>
> Our predictor attaches a newly instantiated linear regression head to a pretrained DistilBERT encoder. The head is initialized with standard PyTorch defaults (zero-centered small-magnitude weights and a near-zero bias), while the encoder’s CLS representation is approximately zero-mean due to LayerNorm.
>
> Consequently, **the initial scalar output $k$ concentrates tightly around zero**. When the continuous regression output is rounded to obtain the discrete speculative step, this yields **$k=0$ consistently at warm-up**.
>
> This behavior constitutes a conservative safety prior rather than instability. As data accrues, the head quickly departs from this prior and k gradually increases toward the empirical value-estimation distribution of current policy.
>
> In addition, our **ablation study explicitly quantifies the warm-up speed of the predictor** (see Appendix A.5 and Table 7). The predictor achieves satisfactory accuracy within 3–5 tasks using a GPT-based backbone and within 8–13 tasks using a DeepSeek-based backbone.
>
> These results confirm that the warm-up process is both **fast and stable**.
>
> **(2) Our prediction decoupled data collection strategy prevents compounding inefficiency** (See line 301-308).
>
> Our training data is collected only when the two agents disagree during verification. This design ensures that **prediction errors affect only system cost and efficiency, without biasing the training data distribution**. Therefore, prediction errors (either over- or under- speculation) do not compound or corrupt the training process.
>
> ---
> > **Q #4. Predictor overhead**
>
> We thank the reviewer for this request for quantification. **Both time and memory overhead of the predictor are negligible**.
>
> **(1) Time: Both training and inference time overhead is negligible compared to the main agent planning pipeline**.
>
> - **Training**: Our predictor training operates with a finite replay buffer. When new training episodes are gathered, we launch a single training session. Each training process iterates over 3 epochs on a randomly sampled batch from the replay buffer (batch size is 16).
> On an A100 GPU, training 3 epochs takes approximately **1.5 second**, which is shorter than a single target agent inference step (GPT-based agents typically take 3 seconds per step; DeepSeek-based agents take over 20 seconds per step) and is **significantly shorter than one speculative episode** (usually take several agent steps).
> - **Inference**: the predictor requires only **~0.05** seconds per prediction inference.
>
> **(2) Memory: The total memory footprint of training and inference predictors combined are still negligible compared to modern GPU memory capacity**.
>
> Our predictor uses DistilBERT-base-uncased as the backbone, which contains approximately **268MB of parameters**.
> We maintain two model copies: one for training and one for inference (updated after each training session) and are still very small.

---

> ### Author Response · Authors · 2025-11-20
> **Response for question 5**
>
> > **Q #5. Hyperparameters Analysis**
>
> We thank the reviewer for this question. We **conducted a comprehensive hyperparameter sensitivity analysis and the hyperparameters reported in the main paper are selected based on the best-performing configurations**.
>
> We now provide a detailed analysis covering **$\lambda$, learning rate, buffer size, batch size, and training frequency**. Among all the hyperparameters, **$\lambda$ and learning rate have the most significant impact on performance**.
>
> Following tables show results of Direct-ReAct configurations with GPT-4.1-mini backbones on OpenAGI benchmarks.
>
> **(1) $\lambda$**
>
> When tuning $\lambda$, we observed the following trend:
> - Smaller $\lambda$ values (e.g., $\leq 0.8$) encourage more aggressive speculation (larger predicted $k$), but often overshoot due to inaccurate predictions and massive cost increase.
> - Larger $\lambda$ values (e.g., $\geq 0.95$) behave more conservatively, with $\lambda = 1.0$ (pure SFT) leading to minimal speedup despite accurate labels.
> This reflects a key tradeoff: **smaller $\lambda$ can amplify prediction noise, while $\lambda = 1.0$ overly adheres to potentially brittle supervision**. In practice, we found that $\lambda \in [0.8, 0.95]$ offers a good balance, enabling value-guided speculation that is both adaptive and stable. Our method uses $\lambda = 0.95$ by default.
>
> **Table 1. Performance Comparison of Different $\lambda$ (First 150 tasks)**
>
> |Mode|$\Delta T$(%)|$\Delta P$(%)|$\Delta G$(%)|$\Delta C$(%)|$\overline{MC}$| $\overline{K}$|
> |---|---|---|---|---|---|---|
> |Fix(k=2)|21.01|27.61|14.78|22.33|3.00|2|
> |Fix(k=4)|28.90|80.41|35.53|60.18|4.93|4|
> |Fix(k=6)|32.23|145.20|64.50|106.28|5.90|6|
> |Dyn($\lambda$=0.1)|23.58|101.70|52.15|77.47|6.02|4.21|
> |Dyn($\lambda$=0.2)|28.81|96.12|49.40|73.59|5.83|4.20|
> |Dyn($\lambda$=0.4)|23.80|64.13|34.33|50.25|5.23|2.62|
> |Dyn($\lambda$=0.6)|24.45|40.94|20.95|32.04|4.85|2.12|
> |Dyn($\lambda$=0.7)|25.24|34.87|18.42|27.54|4.59|1.99|
> |Dyn($\lambda$=0.8)|24.25|27.11|13.13|20.99|4.49|1.89|
> |Dyn($\lambda$=0.95)|23.52|20.60|10.60|16.26|4.46|1.68|
> |Dyn($\lambda$=1.0)|20.54|16.83|8.75|13.41|4.28|1.55|
>
> **(2) Learning rate**
>
> DSP performs well across $lr \in \[1e-6, 1e-4\]$, with **$lr=1e-5$ achieving the best performance**.
>
> **Table 2. Performance Comparison of Different $lr$ (First 200 tasks)**
> |Mode|Time (x)|Prompt (x)|Gen (x)|Cost (x)|MC|
> |---|---|---|---|---|---|
> |Fix (k=2)|1.00|1.00|1.00|1.00|3.00|
> |Fix (k=4)|0.91|1.48|1.21|1.35|4.92|
> |Fix (k=6)|0.88|2.00|1.47|1.74|5.97|
> |Dyn. ($lr=1e-3$)|1.06|1.00|1.01|1.01|4.19|
> |Dyn. ($lr=1e-4$)|0.99|0.95|0.97|0.96|4.69|
> |Dyn. ($lr=1e-5$)|1.00|0.94|0.96|0.94|4.42|
> |Dyn. ($lr=1e-6$)|1.02|0.93|0.97|0.94|4.44|
>
>
> **(3) Buffer size**
>
> A **larger replay buffer (2500) yields the best overall latency–cost performance**, though the differences across 800~2500 remain small.
>
> **Table 3. Performance Comparison of Different Buffer Size (312 whole tasks)**
>
> |Mode|Time (x)|Prompt (x)|Gen (x)|Cost (x)|MC|
> |---|---|---|---|---|---|
> |Fix (k=2)|1.00|1.00|1.00|1.00|3.00|
> |Dyn. (bf=2500)|1.02|0.88|0.92|0.89|4.27|
> |Dyn. (bf=1500)|1.03|0.87|0.91|0.89|4.34|
> |Dyn. (bf=800)|1.04|0.89|0.93|0.91|4.18|
>
> **(4) Batch size**
>
> Batch size **16** delivers the strongest overall performance, though the differences across batch size of 8~32 remain small.
>
> **Table 4. Performance Comparison of Different Batch Size (First 200 tasks)**
>
> |Mode|Time (x)|Prompt (x)|Gen (x)|Cost (x)|MC|
> |---|---|---|---|---|---|
> |Fix (k=2)|1.00|1.00|1.00|1.00|3.00|
> |Dyn. (bs=8)|1.03|0.96|1.00|0.97|4.44|
> |Dyn. (bs=16)|1.00|0.94|0.96|0.94|4.42|
> |Dyn. (bs=32)|1.03|0.95|0.98|0.96|4.59|
>
> **(5) Train frequency**
>
> Train frequency denotes how many newly collected trajectories trigger one predictor-training update.
>
> Training every trajectory (freq = 1) and every three trajectories (freq = 3) produce the best performance.
> We therefore adopt frequency = 1 as the default, without loss of generality.
>
> **Table 5. Performance Comparison of Different Training Frequency (312 whole tasks)**
>
> |Mode|Time (x)|Prompt (x)|Gen (x)|Cost (x)|MC|
> |---|---|---|---|---|---|
> |Fix (k=2)|1.00|1.00|1.00|1.00|3.00|
> |Dyn. (freq=1)|1.02|0.88|0.92|0.89|4.27|
> |Dyn. (freq=2)|1.07|0.85|0.91|0.87|4.12|
> |Dyn. (freq=3)|1.02|0.88|0.92|0.89|4.19|

---

> ### Comment · Reviewer_J1uY · 2025-11-26
>
> Thanks for the detailed response. My questions have largely been solved. I'll adjust the score accordingly.

---

> > ### Author Response · Authors · 2025-11-26
> > **Thank you for raising the score!**
> >
> > Dear reviewer,
> >
> > Thank you very much for your appreciation.
> >
> > Thank you again for raising the score!
> >
> > Best, the authors

---

### Official Review · Reviewer_4PyS · 2025-10-29

**Soundness:** 3
**Presentation:** 2
**Contribution:** 3
**Rating:** 6
**Confidence:** 4

**Summary:**

The authors have created an online RL framework for accelerating LLM-based agents while also being able to lower operational costs. This framework focuses on a limitation of Interactive Speculative Planning. That of a hard-coded speculation depth, which tends to perform badly as the optimal depth can be very different between different tasks and planning stages. The authors have created a speculation step predictor as a state-value estimation problem which is solved with a TD(λ) style algorithm, alongside an asynchronous multi-threaded architecture. Expectile regression and direct offset then allow for fine calibration of the latency-cost tradeoff. Experiments on OpenAGI and TravelPlanner benchmarks show a 30% total cost lowering and up to 60% cost reduction in redundant overhead compared to fixed-depth baselines, with no pre-deployment preparation needed.

**Strengths:**

The problem is clearly well-motivated and highlights an important failing in current approaches. The analysis is rigorous and the demonstration of the sources of redundancy is clear. It's clear why fixed depth doesn't work in general.
The impact shown by the new algorithm also seems very impressive (though perhaps over a small number of tasks), and the fact that there is no pre-deployment preparation is clearly a win.
The multi-threaded architecture itself seems to be well-designed.
The baselines are appropriate (though on first reading I thought that there should have been comparison with System 1.x and ecoact)

**Weaknesses:**

The benchmarks themselves are appropriate but somewhat limited.  They are both domains where it makes sense for the algorithm to do well in, so it would be interesting to see how robust the improvements are in slightly out of distribution domains. For instance, could it be tested with a code-generation agent or an interactive tool-use QA, or indeed just something which is reasoning-heavy but not specifically planning-focused.
It's clearly hard in this domain, but the lack of any statistical analysis is a drawback. Could one apply bootstrapping to get some variance estimates?
There doesn't seem to be any testing for hyper parameter sensitivity for lambda, replay buffer size, batch size or learning rate.
It would be good to quantify the predictor overhead. It says that it's negligible, but it would be useful to know precisely how negligible.
System 1.x and Ecoact are mentioned but it's not explained why these wouldn't be sensible benchmarks to compare against. I see that they are tackling different problems, but it would be useful to have this spelled out explicitly.
Possibly the most important point, and it may be a matter of misunderstanding, but is about the amount of wall-clock time spent exploring the Pareto-frontier. Does that have an overall impact on actually compute time?

**Questions:**

In addition to the above, can you explain if there is a clear way to decide when to choose tau or beta?
When the two agents disagree, how often are those rejections actually harmless?
The DSP relies on running multiple speculative threads in parallel. Have you tested what happens if concurrency is limited? This feels like a real-world constraint that is important to understand.
How much extra wall-clock time does DSP spend exploring the Pareto-frontier?

---

> ### Author Response · Authors · 2025-11-20
> **Response for weakness 1&3&4**
>
> We sincerely appreciate your thorough review and constructive feedback. We are grateful that you acknowledge the strengths of our work, particularly in **the clear motivation, rigorous analysis, impressive empirical results and the novel asynchronous architecture that requires no pre-deployment preparation**.
>
> ---
> > **W #1. Code-generation or Interactive Tool-Use Benchmark**
>
> We appreciate the reviewer’s question. We would like to clarify that code generation tasks often involve irreversible or high-stakes actions, where speculative execution must be applied with greater caution.
>
> However, to **deploy in real life scenarios**, it would require more reliable semantic verification, safety-aware rollback mechanisms, or sophisticated world simulators to prevent false speculative actions from causing irreversible environmental changes or task-level failures, **which the current technical contribution does not focus on and do not contain**.
>
> We view this as an **important and promising direction for future work**, aiming to adapt our dynamic speculative planning framework for safe deployment in interactive and real-world settings.
>
>
> ---
> > **W #3. Predictor overhead**
>
> We thank the reviewer for this request for quantification. **Both time and memory overhead of the predictor are negligible**.
>
> **(1) Time: Both training and inference time overhead is negligible compared to the main agent planning pipeline**.
>
> - **Training**: Our predictor training operates with a finite replay buffer. When new training episodes are gathered, we launch a single training session. Each training process iterates over 3 epochs on a randomly sampled batch from the replay buffer (batch size is 16).
> On an A100 GPU, training 3 epochs takes approximately **1.5 second**, which is shorter than a single target agent inference step (GPT-based agents typically take 3 seconds per step; DeepSeek-based agents take over 20 seconds per step) and is **significantly shorter than one speculative episode** (usually take several agent steps).
> - **Inference**: the predictor requires only **~0.05** seconds per prediction inference, which is much shorter than a single target agent inference step.
>
> **(2) Memory: The total memory footprint of training and inference predictors combined are still negligible compared to modern GPU memory capacity**.
>
> Our predictor uses DistilBERT-base-uncased as the backbone, which contains approximately **268MB of parameters**.
> We maintain two model copies: one for training and one for inference (updated after each training session) and are still very small.
>
> ---
> > **W #4. System 1.x and EcoAct baselines**
>
>
> System-1.x and EcoAct address related efficiency challenges, but **through fundamentally different approaches that require either extensive offline training or intrusive prompt modifications**, whereas DSP achieves adaptive efficiency in a **fully model-agnostic and plug-and-play fashion**.
>
> **(1) System-1.x**: relies on an offline data generation stage that constructs training datasets for three components: a controller, a System-1 planner, and a System-2 planner. This design comes with **significant offline data-collection cost and fine-tuning overhead, and its generalization is bounded by the availability of such search traces.**
>
> In contrast, **DSP performs fully online adaptation on top of pretrained agents without requiring any costly offline data collection or component re-training**. It can be applied to new agent configurations or LLM backbones in a plug-and-play manner, since all learning happens dynamically during task execution rather than through domain-specific fine-tuning.
>
> **(2) EcoAct: requires modifying the agent’s internal prompt and action schema**, by adding a meta-tool (“tool register”) that modifies the agent’s tool-selection process. Moreover, while it reduces context length, this **compression is not lossless**, as it may drop tool descriptions or contextual information that can **affect reasoning fidelity**.

---

> ### Author Response · Authors · 2025-11-20
> **Response for weakness 5 & question 1~3**
>
> > **W #5. Wall-clock time**
>
> Thank you for the helpful suggestions. **DSP quickly stabilizes near the optimal latency-cost region, achieving Pareto-efficient behavior early in training without adding wall-clock cost.**
>
> We provide a performance comparison between DSP and fixed-k baselines on the first 50 and first 100 tasks. (Added in Appendix A12 of the updated pdf).
>
> Within the **first 50 tasks, all DSP configurations except $\tau$ = 0.99 already reach the Pareto frontier**, and **within the first 100 tasks, all DSP variants lie on the frontier**.
>
> In addition, **predictor accuracy serves as an indicator of the online adaptation speed**. As discussed in Appendix A.5, the predictor achieves satisfactory accuracy within 3–5 tasks using a GPT-based backbone and within 8–13 tasks using a DeepSeek-based backbone, demonstrating a **fast and stable warm-up process**. Appendix A.6 further shows a consistent upward trend in mean accuracy with decreasing variance as task experience increases.
>
> > **Q #1. Choice Between Biased Training and Biased Inference**
>
> We thank the reviewer for this important question regarding the selection criteria between $\tau$ (expectile parameter) and $\beta$ (inference-time offset). **We have provided detailed guidance on this decision in Appendix A.4**, where we comprehensively analyze the trade-offs between training-time bias adjustment ($\tau$) and inference-time offset application ($\beta$).
>
> To summarize:
>
> (1) **Training-time bias** (controlled by $\tau$) provides **finer-grained control and better optimization at intermediate points on the efficiency-cost spectrum** and is more suitable when **precise performance calibration** is required.
>
> (2) **Inference-time bias** via a fixed offset parameter $beta$ **offers implementation simplicity, immediate adjustability, and strong performance at discrete operating points, making it suitable when straightforward configuration and rapid adaptation are prioritized.**
>
> Regarding specific trade-off ratios (e.g., how much latency reduction justifies a given cost increase), we found that this value varies significantly across different tasks, deployment scenarios and practitioners’ requirements, making it **impractical to provide a universal quantitative guideline**.
>
> ---
>
> > **Q #2. Harmless Rejections**
>
> We appreciate the reviewer’s insightful question. **We have discussed this limitation in Appendix A.10 (Limitations)**.
>
> Agent planning often admits multiple valid solution paths to accomplish the same task. Our current verification mechanism may reject valid alternative approaches proposed by the approximation agent that differ from the target agent's preferred solution, even when these alternatives could potentially lead to successful task completion. While this conservative verification strategy prioritizes safety and correctness, it does sacrifice some potential acceleration opportunities.
>
> Future work could explore more sophisticated verification strategies to reduce these harmless rejections while maintaining correctness guarantees.
>
> ---
> > **Q #3. Performance under Limited Concurrency**
>
> We thank the reviewer for highlighting this real-world consideration. We have **explicitly evaluated concurrency requirements, and DSP requires lower peak concurrency than fixed-k baselines while achieving better performance**.
>
> (1) Our **evaluation metrics explicitly include $\overline{MC}$ (As discussed in Section 5.1)**, which denotes **average peak concurrency**, i.e. the average maximum number of parallel LLM calls observed per task (see line 398, detailed definition in Appendix A2).
> Average peak concurrency comparisons between different DSP configurations and ISP (fixed k baseline) are presented in our main experimental results (Section 5.1).
>
> Our results show that **DSP achieves lower average peak concurrency compared to fixed k baselines while delivering better performance**, making it more practical for real-world deployment scenarios with concurrency constraints.
>
> (2) **For scenarios with highly limited concurrency**, we recommend using a smaller and faster approximation agent. When $k$ is constrained to be small, we do not expect the approximation agent to generate long sequences of action proposals that perfectly align with the target agent. Thus a smaller and faster approximation agent would be appropriate to deliver comparable acceleration under such constraints.

---

> ### Author Response · Authors · 2025-11-20
> **Response for weakness 2**
>
> > **W #2. Hyperparameters Analysis**
>
> We thank the reviewer for this question. We **conducted a comprehensive hyperparameter sensitivity analysis and the hyperparameters reported in the main paper are selected based on the best-performing configurations**.
>
> We now provide a detailed analysis covering **$\lambda$, learning rate, buffer size, batch size, and training frequency**. Among all the hyperparameters, **$\lambda$ and learning rate have the most significant impact on performance**.
>
> Following tables show results of Direct-ReAct configurations with GPT-4.1-mini backbones on OpenAGI benchmarks.
>
> **(1) $\lambda$**
>
> When tuning $\lambda$, we observed the following trend:
> - Smaller $\lambda$ values (e.g., $\leq 0.8$) encourage more aggressive speculation (larger predicted $k$), but often overshoot due to inaccurate predictions and massive cost increase.
> - Larger $\lambda$ values (e.g., $\geq 0.95$) behave more conservatively, with $\lambda = 1.0$ (pure SFT) leading to minimal speedup despite accurate labels.
> This reflects a key tradeoff: **smaller $\lambda$ can amplify prediction noise, while $\lambda = 1.0$ overly adheres to potentially brittle supervision**. In practice, we found that $\lambda \in [0.8, 0.95]$ offers a good balance, enabling value-guided speculation that is both adaptive and stable. Our method uses $\lambda = 0.95$ by default.
>
> **Table 1. Performance Comparison of Different $\lambda$ (First 150 tasks)**
>
> |Mode|$\Delta T$(%)|$\Delta P$(%)|$\Delta G$(%)|$\Delta C$(%)|$\overline{MC}$| $\overline{K}$|
> |---|---|---|---|---|---|---|
> |Fix(k=2)|21.01|27.61|14.78|22.33|3.00|N/A|
> |Fix(k=4)|28.90|80.41|35.53|60.18|4.93|N/A|
> |Fix(k=6)|32.23|145.20|64.50|106.28|5.90|N/A|
> |Dyn($\lambda$=0.1)|23.58|101.70|52.15|77.47|6.02|4.21|
> |Dyn($\lambda$=0.2)|28.81|96.12|49.40|73.59|5.83|4.20|
> |Dyn($\lambda$=0.4)|23.80|64.13|34.33|50.25|5.23|2.62|
> |Dyn($\lambda$=0.6)|24.45|40.94|20.95|32.04|4.85|2.12|
> |Dyn($\lambda$=0.7)|25.24|34.87|18.42|27.54|4.59|1.99|
> |Dyn($\lambda$=0.8)|24.25|27.11|13.13|20.99|4.49|1.89|
> |Dyn($\lambda$=0.95)|23.52|20.60|10.60|16.26|4.46|1.68|
> |Dyn($\lambda$=1.0)|20.54|16.83|8.75|13.41|4.28|1.55|
>
> **(2) Learning rate**
>
> DSP performs well across $lr \in \[1e-6, 1e-4\]$, with **$lr=1e-5$ achieving the best performance**.
>
> **Table 2. Performance Comparison of Different $lr$ (First 200 tasks)**
> |Mode|Time (x)|Prompt (x)|Gen (x)|Cost (x)|MC|
> |---|---|---|---|---|---|
> |Fix (k=2)|1.00|1.00|1.00|1.00|3.00|
> |Fix (k=4)|0.91|1.48|1.21|1.35|4.92|
> |Fix (k=6)|0.88|2.00|1.47|1.74|5.97|
> |Dyn. ($lr=1e-3$)|1.06|1.00|1.01|1.01|4.19|
> |Dyn. ($lr=1e-4$)|0.99|0.95|0.97|0.96|4.69|
> |Dyn. ($lr=1e-5$)|1.00|0.94|0.96|0.94|4.42|
> |Dyn. ($lr=1e-6$)|1.02|0.93|0.97|0.94|4.44|
>
>
> **(3) Buffer size**
>
> A **larger replay buffer (2500) yields the best overall latency–cost performance**, though the differences across 800–2500 remain small.
>
> **Table 3. Performance Comparison of Different Buffer Size (312 whole tasks)**
>
> |Mode|Time (x)|Prompt (x)|Gen (x)|Cost (x)|MC|
> |---|---|---|---|---|---|
> |Fix (k=2)|1.00|1.00|1.00|1.00|3.00|
> |Dyn. (bf=2500)|1.02|0.88|0.92|0.89|4.27|
> |Dyn. (bf=1500)|1.03|0.87|0.91|0.89|4.34|
> |Dyn. (bf=800)|1.04|0.89|0.93|0.91|4.18|
>
> **(4) Batch size**
>
> Batch size **16** delivers the strongest overall performance.
>
> **Table 4. Performance Comparison of Different Batch Size (First 200 tasks)**
>
> |Mode|Time (x)|Prompt (x)|Gen (x)|Cost (x)|MC|
> |---|---|---|---|---|---|
> |Fix (k=2)|1.00|1.00|1.00|1.00|3.00|
> |Dyn. (bs=8)|1.03|0.96|1.00|0.97|4.44|
> |Dyn. (bs=16)|1.00|0.94|0.96|0.94|4.42|
> |Dyn. (bs=32)|1.03|0.95|0.98|0.96|4.59|
>
> **(5) Train frequency**
>
> Train frequency denotes how many newly collected trajectories trigger one predictor-training update.
>
> Training every trajectory (freq = 1) and every three trajectories (freq = 3) produce the best performance.
> We therefore adopt frequency = 1 as the default, without loss of generality.
>
> **Table 5. Performance Comparison of Different Training Frequency (312 whole tasks)**
>
> |Mode|Time (x)|Prompt (x)|Gen (x)|Cost (x)|MC|
> |---|---|---|---|---|---|
> |Fix (k=2)|1.00|1.00|1.00|1.00|3.00|
> |Dyn. (freq=1)|1.02|0.88|0.92|0.89|4.27|
> |Dyn. (freq=2)|1.07|0.85|0.91|0.87|4.12|
> |Dyn. (freq=3)|1.02|0.88|0.92|0.89|4.19|

---

> > ### Comment · Reviewer_4PyS · 2025-11-22
> > **Response to author's responses**
> >
> > I thank the authors for going through each of my concerns carefully. I am happy with these responses, some of which are useful additions and some of which have clarified misunderstandings. I will adjust my score accordingly.

---

> > > ### Author Response · Authors · 2025-11-26
> > > **Thank you for raising the score!**
> > >
> > > Dear reviewer,
> > >
> > > Thank you very much for your appreciation.
> > >
> > > Thank you again for raising the score!
> > >
> > > Best, the authors

---

### Official Review · Reviewer_suqa · 2025-11-01

**Soundness:** 3
**Presentation:** 3
**Contribution:** 3
**Rating:** 6
**Confidence:** 3

**Summary:**

This paper proposes a reinforcement learning–based framework to accelerate large language model (LLM) agents while maintaining their performance quality. Traditional acceleration techniques often require expensive pretraining or sacrifice accuracy; DSP instead introduces a lightweight, online learning mechanism that dynamically adjusts how far ahead a “draft” agent speculates before verification by a “target” agent. Using asynchronous temporal-difference learning, DSP continually refines its prediction of optimal speculation steps during live operation, requiring no pre-deployment setup. Moreover, it offers user-controllable tradeoffs between cost and latency through expectile regression or simple offset parameters. Experiments on OpenAGI and TravelPlanner benchmarks show that DSP maintains comparable or better acceleration to existing lossless methods while cutting total cost by up to 30% and eliminating 60% of redundant computation, making it an efficient and flexible solution for real-world LLM-based agents.

**Strengths:**

1. The paper uses online reinforcement learning (TD learning) to adjust the speculation step k without pretraining, achieving lossless acceleration while maintaining performance.
2. It provides user-controllable tradeoffs between speed and cost through expectile regression and offset parameters.
3. The work demonstrates strong empirical results with up to 30% total cost reduction and 60% reduction in redundant cost.

**Weaknesses:**

1. The method’s technical novelty is incremental, as it mainly extends existing speculative planning with an adaptive step predictor.
2. The online RL part may introduce instability or convergence issues in dynamic, real-world settings. More specifically, the real-world LLM applications could have countless new scenarios, and it can be hard to make sure that the RL training has covered enough state-action spaces.
3. DSP’s performance depends on the quality of the initial predictor, which might require some warm-up time before optimal behavior emerges.

**Questions:**

1. How do you plan to measure the generalization gap, and is there any safety net for out-of-distribution state-action pairs at test time?

2. Will setting gamma to 1 cause instability for RL training? Why not use a more theoretically sound number like .99?

---

> ### Author Response · Authors · 2025-11-20
> **Response for question 1&2**
>
> We sincerely appreciate your constructive review. We're encouraged that you recognized our use of **online RL framework, the practical controllability of speed-cost tradeoffs and strong empirical gains**.
>
>
> ---
>
> > **Q #1.  Generalization Gap and OOD Safety Mechanisms**
>
> **(1) Generalizability:**
>
> **The online learning process itself and the accuracy of the predictor directly measure generalization.**
>
> Our predictor is trained from scratch in an online fashion. It starts from random initialization and learns as it sequentially processes each task in the benchmark. Each task in these benchmarks is unique and unseen at training, with no overlap or data leakage.
>
> Consequently, the **predictor’s accuracy directly reflects its generalization to new, unseen task distributions**. As shown in Appendix A.6, the predictor exhibits a clear upward trajectory in mean accuracy with decreasing variance, indicating that it generalizes its value estimation across diverse prompts and contexts rather than memorizing specific instances.
>
> **(2) Safety Mechanisms:**
> - **Empirical prediction stability**: we do not observe harmful out-of-distribution predictions under our standard hyperparameters. The predictor remains stable across all settings, and we have not observed any unusually large or abnormal $k$ predictions.
> - **Stability under distribution shift**: To further stress-test distribution shift, we trained the predictor online on all OpenAGI tasks and directly applied the resulting checkpoint to TravelPlanner. Despite domain differences, the predictor immediately produced reasonable $k$ values (99\% within {1,2}) and maintained good accuracy, indicating strong robustness to unseen state–action distributions.
>
>
> - **Safety Net is easy to implement**: You are absolutely right to ask about the safety net! If additional safeguards are desired, simple heuristics can be easily implemented. For example, we could detect when the distance between consecutive tasks exceeds a threshold and heuristically cap the predicted k value. However, developing safety nets is not the primary technical contribution of this work. But we definitely encourage people to implement this when deploy DSP.
>
> **Table 1. Predictor Accuracy on first 14 tasks of TravelPlanner**
> |Task|1|2|3|4|5|6|7|8|9|10|11|12|13|14|
> |-|-|-|-|-|-|-|-|-|-|-|-|-|-|-|
> |Accuracy|0.86|0.62|0.62|0.75|0.57|0.62|0.43|0.57|0.67|1.0|0.6|0.71|1.0|0.62|
>
> ---
>
>
> > **Q #2. Discount Factor Choice and Training Stability**
>
> **Setting $\gamma$=1 will not cause instability in our system**.
>
> In fact, **$\gamma$=1 offers both theoretical advantages in interpretability and empirical improvements compared to $\gamma$=0.99.**
>
> **(1) Theoretically**: We set $\gamma=1$ so that the **return $G_t$ directly corresponds to the total number of speculative steps at time-step $t$, preserving a clear, interpretable metric**. While $\gamma<1$ (e.g., 0.99) is theoretically common, it would obscure this direct correspondence. To mitigate potential instability, we further employ $\lambda$-returns, which provide a principled bias-variance trade-off[1].
>
> **(2) Empirically**: We conducted an additional experiment comparing $\gamma$=1.0 and $\gamma$=0.99 on the OpenAGI benchmark with GPT-4.1-backbone (Direct-ReAct configuration). The results show no significant performance difference between the two settings, while **$\gamma$=1.0 achieves better average maximum concurrency**:
>
> **Table 2. Performance Comparison Between Different $\gamma$**
>
> |Method|Time (x)|Prompt (x)|Gen (x)|Cost (x)|$\overline{MC}$|
> |---|---|---|---|---|---|
> |Dyn. ($\gamma$=1.0, $\tau$=0.5)|1.03|0.87|0.91|0.88|4.33|
> |Dyn. ($\gamma$=0.99, $\tau$=0.5)|1.03|0.86|0.91|0.88|4.79|
>
> [1] "Proximal policy optimization algorithms." arXiv preprint arXiv:1707.06347 (2017)

---

> ### Author Response · Authors · 2025-11-20
> **Response for question 3**
>
> > **Q #3. Predictor Warm-Up and Its Impact on DSP Stability**
>
> We would like to clarify that **all reported results include performance starting from task 0**. In addition, the predictor’s initial behavior is **inherently cost-safe (outputting $k=0$) with a very fast warm-up**.
>
> **(1) The predictor consistently outputs k=0 during initialization, and this conservative behavior prevents instability**.
>
> This is because our predictor attaches a newly instantiated linear regression head to a pretrained DistilBERT encoder. The head is initialized with standard PyTorch defaults (zero-centered small-magnitude weights and a near-zero bias), while the encoder’s CLS representation is approximately zero-mean due to LayerNorm.
>
> Consequently, the initial scalar output k concentrates tightly around zero, **yielding k=0 consistently at warm-up** when rounded to discrete values. This behavior constitutes a conservative safety prior rather than instability. As data accrues, the head quickly departs from this prior and k gradually increases toward the empirical value-estimation distribution of current policy.
>
> **(2) In addition, our ablation study explicitly quantifies the warm-up speed of the predictor** (see Appendix A.5 and Table 7).
>
> With a **GPT-based backbone**, the predictor typically achieves satisfactory prediction accuracy within **3–5 tasks**, while with a **DeepSeek backbone**, this can be achieved within **8–13 tasks**.
>
> These results confirm that the warm-up process is both **fast and stable**.

---

> > ### Comment · Reviewer_suqa · 2025-11-26
> >
> > Thanks for your detailed response. My questions have been solved. I'll maintain my positive score.

---

> > > ### Author Response · Authors · 2025-11-26
> > > **Further discussion and kind request for potential score increase**
> > >
> > > Dear Reviewer suqa,
> > >
> > > We sincerely appreciate the time and effort you've invested in reviewing our paper. Your insightful comments and thoughtful feedback are invaluable to us.
> > >
> > > We hope that our responses have addressed your questions and clarified any ambiguities. Please do not hesitate to reach out if you have any further questions!
> > >
> > > **We would be truly grateful if you would consider our clarifications and kindly reevaluate your score.**
> > >
> > > Thank you once again for your consideration.
> > >
> > > Best regards, the authors

---

### Official Review · Reviewer_QcJ4 · 2025-11-03

**Soundness:** 2
**Presentation:** 3
**Contribution:** 2
**Rating:** 6
**Confidence:** 3

**Summary:**

The work proposes a method (DSP) where they use a target LLM and a smaller approximation LLM to accelerate multi-step agent planning through speculative execution. The approximation LLM speculatively generates multiple future planning steps ahead in parallel, while the target LLM simultaneously verifies each step. When the two agents agree, steps are committed to the final plan. When they disagree, the incorrect speculative steps are discarded and execution resumes from the target LLM's decision. The main contribution of this work is dynamically predicting how many steps ahead to speculate using an online-trained distillbert predictor based on TD learning, rather than using a fixed speculation depth. This prediction is adjusted using either expectile regression or a biased offset to let users control the latency-cost tradeoff. Results show they achieve comparable acceleration to aggressive fixed-depth approaches while reducing both total costs and unnecessary computational waste.

**Strengths:**

- DSP achieves Pareto dominance across multiple agent configurations and LLM families, delivering comparable acceleration to aggressive fixed-depth approaches while reducing costs.
- The ablations are well done.
- Dynamic speculation depth via online RL is seems reasonably novel compared to fixed-k prior work.

**Weaknesses:**

- Only OpenAGI and TravelPlanner are evaluated. They for instance mention the applicability of the work in software engineering but do not have any results on any such benchmark.
- I am not sure if authors evaluate on multiple seeds or not, given the online RL part, I believe multiple seeds are extremely important to establish statistical relevance.
- Authors do not discuss why TravelPlanner shows substantially weaker improvements than OpenAGI (7 percent cost reduction vs. 30 percent), some discussion around that will be good to have.
- System 1.x (Saha et al. 2025) and EcoAct (Zhang et al. 2024b) are cited as addressing similar latency challenges but are not compared.

**Questions:**

- You claim applicability to software engineering  too, but there is no result on such a benchmark (like SWE bench), can you explain why?
- Were experiments run with multiple seeds?
- Why does DSP achieve 30 percent cost reduction on OpenAGI but only 7 percent on TravelPlanner? What task characteristics drive this variability?
- System 1.x and EcoAct are cited as addressing similar latency challenges but not compared experimentally. Why were they excluded?

---

> ### Author Response · Authors · 2025-11-20
> **Response for weakness 1 & 2**
>
> We sincerely appreciate your thoughtful review and constructive feedback. We are especially grateful that you highlighted the strengths of our work, including **DSP's Pareto dominance across different settings, the comprehensiveness of ablation studies, and the novelty of online RL framework**.
>
> ---
>
> **W #1. Software Engineering Benchmark**
>
> We thank the reviewer’s attention to this point. The mention of “autonomous software engineering” in the introduction was intended to **illustrate the broader context of LLM-based agent applications, not to imply that DSP was directly evaluated on SWE-bench or similar datasets**.
>
> We would also like to clarify that while DSP’s online learning and plug-and-play design conceptually allow application to software-engineering agents, it often involve **irreversible or high-stakes actions** (e.g., code execution or modification), where speculative planning must be applied with greater caution.
>
> However, to **deploy in real life scenarios**, it would require more reliable semantic verification, safety-aware rollback mechanisms, or sophisticated world simulators to prevent false speculative actions from causing irreversible environmental changes or task-level failures, **which the current technical contribution does not focus on and do not contain**.
>
> We view this as an **important and promising direction for future work**, aiming to adapt our dynamic speculative planning framework for safe deployment in interactive and real-world settings.
>
> ---
>
> **W #2. Multiple Seeds Experimental Results**
>
> **DSP's performance and Pareto dominance are robust across different random seeds**. Our main experiments were conducted using seed 2. We additionally ran the Direct-ReAct configuration with the GPT-4.1-mini backbone on the OpenAGI benchmark under two different random seed 42 and random seed 0. The full results are shown below:
> This confirms that our key findings are stable and not sensitive to seed choice.
>
> **Table 1(a). Performance Comparison (Seed 42)**
>
> |Method|Time(x)|Prompt(x)|Gen(x)|Cost(x)|$\overline{MC}$|
> |---|---|---|---|---|---|
> |Fix(k=2)|1.00|1.00|1.00|1.00|3.00|
> |Fix(k=4)|0.92|1.56|1.23|1.41|4.95|
> |Fix(k=6)|0.90|2.24|1.57|1.92|6.32|
> |Dyn.($\tau$=0.5)|1.01|0.87|0.92|0.89|4.52|
> |Dyn.($\tau$=0.8)|0.98|0.97|0.99|0.97|4.99|
> |Dyn.($\tau$=0.9)|0.94|1.06|1.05|1.05|5.30|
> |Dyn.($\tau$=0.95)|0.94|1.13|1.10|1.12|5.46|
> |Dyn.($\tau$=0.99)|0.91|1.29|1.19|1.24|5.82|
>
> **Table 1(b). Performance Comparison (Seed 2)**
>
> |Method|Time(x)|Prompt(x)|Gen(x)|Cost(x)|$\overline{MC}$|
> |---|---|---|---|---|---|
> |Dyn.($\tau$=0.5)|1.03|0.87|0.91|0.88|4.33|
> |Dyn.($\tau$=0.8)|0.98|0.97|0.98|0.97|4.41|
> |Dyn.($\tau$=0.9)|0.96|1.04|1.00|1.02|4.56|
> |Dyn.($\tau$=0.95)|0.95|1.12|1.07|1.09|4.75|
> |Dyn.($\tau$=0.99)|0.92|1.32|1.19|1.25|5.24|
>
> **Table 1(c). Performance Comparison (Seed 0)**
>
> |Mode|Time (x)|Prompt (x)|Gen (x)|Cost (x)|$\overline{MC}$|
> |---|---|---|---|---|---|
> |Dyn. ($\tau$=0.5)|1.02|0.91|0.96|0.93|4.65|
> |Dyn. ($\tau$=0.8)|0.96|0.96|1.01|0.98|5.00|
> |Dyn. ($\tau$=0.9)|0.94|1.02|1.02|1.02|5.24|
> |Dyn. ($\tau$=0.95)|0.94|1.14|1.12|1.13|5.41|
> |Dyn. ($\tau$=0.99)|0.90|1.33|1.23|1.28|6.00|
>
> Table 1(b)(c) share the same fix baselines with Table 1(a).

---

> ### Author Response · Authors · 2025-11-20
> **Response for weakness 3 & 4**
>
> > **W #3. Performance comparisons between TravelPlanner and OpenAGI benchmark**
>
> Thank you for raising this point. We would like to clarify the **reported percentages reflect different baseline choices**.
>
> The **7\% represents improvement over the most cost-efficient baseline of TravelPlanner**. However, when compared to the **fastest fixed baseline**, DSP achieves substantial cost reductions on both benchmarks: **30\% on OpenAGI and 24.39\% on TravelPlanner**.
>
> The 7\% cost reduction is achieved in Setting 1 (Direct-ReAct agent pairs) compared with the most cost-efficient baseline, which is to apply ISP with minimal k (Fix($k=2$)) which costs the least with the least non-trivial speedup: comparing with this baseline, DSP still reduces total cost even less than k = 2 by 7\% (17\% redundant cost reduction) with better speedup.
>
> When comparing with **the fastest baseline**, which is to may k as high as possible (here k = 6) (in setting 2), DSP($\tau=0.99$) **reduce total cost by a significant 24.39\% (42\% reduction in redundant costs)**, even with better speedup.
>
>
> ---
>
> > **W #4. System 1.x and EcoAct baselines**
>
>
> System-1.x and EcoAct address related efficiency challenges, but **through fundamentally different approaches that require either extensive offline training or intrusive prompt modifications**, whereas DSP achieves adaptive efficiency in a **fully model-agnostic and plug-and-play fashion**.
>
> **(1) System-1.x**: relies on an offline data generation stage that constructs training datasets for three components: a controller, a System-1 planner, and a System-2 planner. This design comes with **significant offline data-collection cost and fine-tuning overhead, and its generalization is bounded by the availability of such search traces.**
> In contrast, **DSP performs fully online adaptation on top of pretrained agents without requiring any costly offline data collection or component re-training**. It can be applied to new agent configurations or LLM backbones in a plug-and-play manner, since all learning happens dynamically during task execution rather than through domain-specific fine-tuning.
>
> **(2) EcoAct: requires modifying the agent’s internal prompt and action schema**, by adding a meta-tool (“tool register”) that modifies the agent’s tool-selection process. Moreover, while it reduces context length, this **compression is not lossless**, as it may drop tool descriptions or contextual information that can **affect reasoning fidelity**.

---

> ### Comment · Reviewer_QcJ4 · 2025-11-22
> **Response to authors**
>
> Thank you for your rebuttal, these are my views:
>
> 1. I am satisfied with the per-seed stability results.
> 2. I am also somewhat satisfied with the arguments presented for 3 & 4 (Please ignore the first edit, I overlooked the response to 3&4, I am more or less satisfied now.)
>
> Therefore, I will keep my score as it is but I am more confident of my decision.

---

### Comment · Area_Chair_kVPk · 2025-11-25
**Please read the rebuttal and respond**

Dear reviewers,

Now that the author responses are in, could you please take a look at them and see if they address your concerns adequately?

Thank you very much.

Best,
AC

---

### Author Response · Authors · 2025-12-01
**General Response**

Dear AC, SAC, and PC:

We sincerely appreciate your time and dedication to the ICLR community and the peer-review process. To reduce your reviewing burden and further highlight the contributions of our paper while clarifying its limitations, we summarize our responses as follows:

---

### **Core Contribution**

**1.Dynamic Speculative Planning (DSP) as a general-purpose, plug-and-play mechanism.**
DSP introduces a fully online, model-agnostic framework that adaptively selects speculation depth for multi-step agent planning.

**2.Online TD(λ)-based value estimation enabling real-time adaptability.**
DSP reframes speculation depth as a value-estimation problem and uses online TD(λ) to update this value in real time. This turns speculative planning from a fixed heuristic into a principled, self-correcting control mechanism that consistently outperforms fixed-k baselines across models and settings.

**3.Controllable latency–cost tradeoff.**
DSP supports fine-grained and stable control over performance, allowing practitioners to navigate the usability frontier (latency vs. cost) through a single parameter.

**4.Strong and consistent Pareto improvements across tasks, seeds, and settings.**
DSP achieves up to 30\% reduction in total cost and 60\% reduction in redundant computation, dominating fixed-k baselines under both latency and cost metrics while also requiring lower peak concurrency.

---

### **Strengths Recognized by Reviewers**
- Clear motivation and important problem
(`QcJ4`, `4PyS`, `J1uY`)

- Novel and effective online RL framework
(`suqa`, `4PyS`, `J1uY`)

- Strong empirical results and Pareto improvements
(`QcJ4`, `suqa`, `4PyS`, `J1uY`)

- Thorough ablations and rigorous analysis
(`QcJ4`, `4PyS`, `J1uY`)

- Practical, plug-and-play design requiring no pre-deployment preparation
(`suqa`, `4PyS`, `J1uY`)

---
### **Reviewer Main Concerns & Our Responses**
**1. System-1.x and EcoAct baselines:**
We clarified that System-1.x and EcoAct address related efficiency challenges, but through fundamentally different approaches that require either expensive offline data gathering and training or intrusive prompt modifications, whereas DSP achieves adaptive efficiency in a fully model-agnostic and plug-and-play fashion. (`QcJ4`, `J1uY`, `4PyS`)

**2. Hyperparameter sensitivity:**
We presented a comprehensive sensitivity study covering different $\lambda$, learning rates, replay buffers, batch sizes and training frequencies. Among all the hyperparameters, $\lambda$ and learning rate have the most significant impact on performance. The hyperparameters reported in the main paper are selected based on the best-performing configurations. (`J1uY`, `4PyS`)


**3. Predictor overhead:** We provided exact measurements of time and memory overhead of the predictor. Both training(\~1.5s per update, much shorter than a speculative planning episode) and inference(\~0.05s) time overhead is negligible compared to the main agent planning pipeline. Parameters of two DistilBERT copies (\~268MB each) are negligible compared to modern GPU capacity. (`J1uY`, `4PyS`)


**4. Predictor's behavior during warm-up phase:** We clarified that the predictor’s warm-up dynamics are **inherently stable and cost efficient**. The predictor initializes conservatively at $k=0$ due to zero-centered weights and LayerNorm, which prevents instability and oscillation behavior during the warm-up phase. (`suqa`, `J1uY`)

---
### **Reviewer Feedback Summary**
All reviewers agreed that the rebuttal fully or largely addressed their concerns.
| Reviewer | Post-Rebuttal Statement | Reply Date|
 | --- | --- |---|
| **`4PyS`** | **"I am happy with these responses... I will adjust my score accordingly."** | 22 Nov 2025|
| **`J1uY`** | **"My questions have largely been solved. I'll adjust the score accordingly."** |26 Nov 2025|
| **`QcJ4`** | **"I am satisfied... I am more confident of my decision."** | 22 Nov 2025|
| **`suqa`** | **"My questions have been solved. I'll maintain my positive score."** | 25 Nov 2025|

To address the concern of reviewer `4PyS`, we also added text and figures in Appendix A.12 (highlighted in blue) to present wall-clock time analysis of DSP in the revised PDF (uploaded Nov 20).

We respectfully submit this summary for your consideration. Thank you again for your time and efforts on our work.

Best regards,

Authors

---

### Meta-Review · Area_Chair_xvGg · 2026-01-06

**Summary:**

This work studies the acceleration of LLM-based agent planning through speculative planning. The core contribution is the introduction of a predictor that adaptively determines the speculative planning length based on the current state. The main concerns raised by reviewers relate to the limited set of benchmarks and the stability of the predictor during the warm-up phase. I would recommend that the authors conduct additional experiments beyond the current scope to better demonstrate the generality of their algorithm.

**Reviewer Concerns:**

For simplicity, I list all concerns that I believe have not been fully addressed by the authors.

**Reviewer QcJ4:** lack of experiments on broader benchmarks, such as SWE-bench.

**Reviewer suqa:** limited novelty, and insufficient justification of the stability of the proposed online RL approach.

**Reviewer 4PyS:** limited benchmark coverage.

**Reviewer J1uY:** limited benchmark coverage.

**Reviewer Scores:**

**Reviewer QcJ4:** 6 → 6

**Reviewer suqa:** 6 → 6

**Reviewer 4PyS:** 6 → 8

**Reviewer J1uY:** 4 → 6

---

### Decision · Program_Chairs · 2026-01-26

Accept (Poster)